# NOISY QUADRATIC MODELS OF SCALING DYNAMICS

## ABSTRACT

Pre-training scaling laws describe the best training decisions under resource constraints. The discovery of new laws is a demanding exercise, as each decision requires a separate law. An alternative is to model the scaling dynamics of LLMs directly, then use those models as surrogates for multiple decisions. Yet, most theoretical models of scaling dynamics cannot be fit to scaling data easily. In this paper, we introduce the *Noisy Quadratic System (NQS)*, a fittable relative of the theoretical models that can generate new scaling laws. We also identify some key failure modes in the theoretical models, and further extend the NQS to correct for these deficiencies. In our experiments, our best model, fit on small-scale runs, closely predicted the performance of runs near critical points, which Chinchilla failed to do. Finally, the NQS is the first practical scaling model to include a variance term, which allows us to model the effect of batch size. In our experiments, we show how to use the NQS to decide batch size, training steps, and model size under many resource constraints, including compute, but also time and memory.

## 1 INTRODUCTION (REVISED)

Pre-training scaling laws describe how the performance of large language models (LLM) improve predictably with increasing resources. The cross-entropy of well-trained LLMs follows a power-law relationship with training compute (Achiam et al., 2024).

Some of these laws prescribe training recipes that would make best use of the resources. For example, Hoffmann et al. (2022) found that the optimal model size is a function of the total compute; DeepSeek LLM used power laws to predict optimal batch sizes from training compute (Bi et al., 2024); and more recently, Bergsma et al. (2025) showed that a power law in dataset size more closely predicts the optimal batch size. This progress is critical, but each training decision required a separate law, each with extensive experiments, bespoke heuristic arguments, and clever insights.

An alternative to scaling laws is to build a single model that predicts LLM test loss as a function of all relevant training configurations; the model acts as a surrogate of actual pre-training, and decisions can be made by minizing the predicted loss. We refer to such approaches as "a scaling model" (in contrast to "scaling laws"). Approach 3 in Hoffmann et al. (2022) is a step in this direction: they fit a two-term power law to predict LLM test loss from model and dataset size. However, the model does not incorporate important pre-training decisions like batch size, learning rate, optimizers and schedules.

In this paper we introduce a richer scaling model class, called the *Noisy Quadratic System (NQS)*. In particular, we can model batch size and batch size schedules, which allows one to allocate memory and time (both functions of batch size) and select the optimal schedules.

The NQS is derived by gathering and simplifying the assumptions in prior works, including statistical models from the theory literature, and the Noisy Quadratic Model. We side-stepped the challenges of the theoretical analysis using numerical methods and recursions, making these models efficiently computable and fittable to scaling data. We also identified a couple of areas where LLM training dynamics significantly deviate from quadratic models, and extended the NQS to correct for the gap.

The extended model, called NQS$^{++}$, closely tracked the behavior of LLM losses across batch sizes, explaining $\geq 90\%$ of the variations due to token allocation on out-of-sample token budgets. The NQS$^{++}$ consistently chose configurations that were close to the ground truth optimal. NQS$^{++}$

also showed promise for high-dimensional configurations, correctly ranking a number of batch size schedules over a range of average batch sizes.

Because the NQS model class is more elaborate than Chinchilla, we need to make sure that positive results are not from overfitting. We used the standard statistical approach to compare models of varying complexity: dividing the data into training/validation/test split, fitting the model on the training split, and assessing the models on the test set. To make the analysis more relevant to scaling analysis, LLMs in the test split is up to x64 larger than those in training, and the test split is only revealed once NQS$^{++}$is fully developed. We found no evidence of NQS$^{++}$overfitting.

Surprisingly, NQS$^{++}$is more robust than the apparently simpler Chinchilla model: because the NQS is highly structured, it may contain beneficial inductive bias. In our experiments, NQS$^{++}$ was able to reproduce Chinchilla scaling laws and robustly extrapolate over compute scales, explaining $\geq 85\%$ of the variation due to $N, D$ allocation out-of-sample. In contrast, Chinchilla overfitted on training data, not only on our smaller scale LLM dataset, but also on its original Hoffman dataset (see E.5).

As a bridge between theory and practice, the NQS is not yet fully mechanistic, and does not model some important pre-training configurations like the learning rate; however, our rolled-out optimization approach makes the NQS easy to extend and modify, and no clever insights were required to apply the model to new tasks. We hope as the NQS model class matures, it would offer a consolidated solution to pre-training decisions.

## 2   A BACKGROUND IN SCALING MODELS (REVISED)

In this section, we briefly review the existing scaling laws and scaling model. We took inspirations from these work to develop NQS$^{++}$. For a complete discussion, please refer to App. A.

### 2.1   SCALING MODELS

A scaling model predicts the test loss of an LLM using its pre-training configurations. Specifically, let $L_i^{\mathrm{LLM}}$ be a test loss obtained after training an LLM from a certain *model family*, configured to use $N_i$ trainable parameters and $K_i$ steps of an optimization algorithm with batch size $B_i$. We define a model family as a mapping from a model size $N$ to a complete trainable LLM architecture (see F.1 for an example)[1].

Denote the test loss $L^{\mathrm{LLM}}$ for new configurations $(N, B, K)$ as $N, B, K \to \infty$. We predict this value via a parametric model $L_\theta^{\mathrm{SM}}$ that minimizes an empirical loss over a "training" subset of *scaling data* $(N_i, B_i, K_i, L_i^{\mathrm{LLM}})$,

$$\theta^* = \underset{\theta}{\mathrm{argmin}} \, \frac{1}{|\mathrm{train}|} \sum_{i \in \mathrm{train}} \mathcal{L}(L_\theta^{\mathrm{SM}}(N_i, B_i, K_i), L_i). \tag{1}$$

In our experiments, we took $\mathcal{L}$ to be the Huber loss between the logarithms of its two arguments, as in Hoffmann et al. (2022). We refer to models $L_\theta^{\mathrm{SM}}$ as *scaling models*.

> **A Practical Scaling Model: Chinchilla Approach 3 (Hoffmann et al., 2022)**
>
> The **Chinchilla Approach** (Hoffmann et al., 2022) is the most widely used scaling model:
>
> $$L_\theta^{\mathrm{CHIN}}(N, D) = \mathcal{E}_{\mathrm{irr}} + \frac{P}{N^{p-1}} + \frac{Q}{D^{p/q-1/q}}, \tag{2}$$
>
> where $D = B \times K \times$ seq. length is the total number of tokens used in training, and $\theta = (p, P, q, Q, \mathcal{E}_{\mathrm{irr}}) \in (\mathbb{R}^{\geq 0})^5$ are scaling parameters satisfying $p > 1$.[a]
>
> ---
> [a]We re-parameterized Chinchilla for consistency with our scaling model's semantics.

Scaling models can be used to allocate resources or select configurations. E.g., suppose you have a GPU with $m$ MB of vRAM and your energy limit affords you at most $c$ floating point operations

---
[1]Note that architectural choices influence the scaling of quantities like compute. We assume knowledge of the model family when calculating these quantities throughout.

Table 1: Definitions used throughout. Where clear, we suppress the dependence on the configuration.

| Configuration Quantities | $N$ | Model size: number of trainable parameters |
| | $B$ | Batch size: examples (or tokens) per optimizer step |
| | $K$ | Training steps: number of optimizer updates (iterations) |
| Resource Quantities | $D(B, K)$ | Dataset size: number of training tokens = $B \times K \times$ seq. length |
| | $C(N, B, K)$ | Training FLOPs: training compute = $6 \times N \times D$ |
| | $M(N, B)$ | Peak memory: peak GPU memory (MB) (Rees (2023)) |

(FLOPs). If $L_{\theta^*}^{\mathrm{SM}} \approx L^{\mathrm{LLM}}$, then you can use it as a surrogate to determine the best model size, batch size, and training steps subject to a constraint on FLOPs $C(N, B, K)$ and peak memory $M(N, B)$,

$$N^*(c,m), B^*(c,m), K^*(c,m) = \underset{\substack{C(N,B,K) \leq c \\ M(N,B) \leq m}}{\operatorname{argmin}} L_{\theta^*}^{\mathrm{SM}}(N, B, K). \tag{3}$$

Generalizations of eq. (3) can be used to allocate other resources. table 1 summarizes our notation.

## 2.2 THEORETICAL SCALING MODELS

Statistical models that qualitatively match LLM scaling dynamics are proposed by theorists (e.g., Bahri et al., 2021; Maloney et al., 2022). For a high-level understanding, we present informally the assumptions in Maloney et al. (2022), a representative of the linear regression family of such models. Later, when we introduce NQS, we described how these assumptions can be stated precisely and/or transformed for practical use.

---

**Assumptions in the Linear Regression Theoretical Scaling Models (Maloney et al., 2022)**

**Data Generation:** A very high-dimensional latent input space $X$ with covariance eigenvalues following a power-law distribution. Labels $y$ are generated from the latent space via a high-dimensional linear map, $y = w^* X + \epsilon$, where $w^*$ has zero-mean random coefficients with power-law covariance, and $\epsilon$ is additive noise.

**Modeling:** The modeler does not observe $X$ directly. Instead, a lower dimensional variable $\Phi$ is given: $\Phi = XP$, where $P$ is a random Gaussian matrix mapping $X$ into a lower-dimensional "feature" space. The modeler's objective is to recover $y$ using $\Phi$ via linear regression $y \sim w^{\mathrm{est}} \Phi$. Maloney et al. (2022) solved the regression problem via ERM, while later work uses gradient flow (Bordelon et al., 2024) or SGD (Paquette et al., 2025).

**Analogy to LLM pre-training:** The expected risk of $y \sim w^{est} \Phi$ is a *quadratic function* in the parameters $w^{est}$ ($\mathcal{Q}(w^{est})$), and can be broken down into the irreducible error (due to noise in $y$), the approximation error (since the random feature projection removed dimensions, the model can never be perfect), and bias, variance (similar to standard linear regression, these two terms are functions of the optimization algorithm).

$$\text{Expected Risk} = \mathcal{Q}(w^{est}) = \mathcal{E}_{irr} + \mathcal{E}_{appx} + \mathcal{E}_{bias} + \mathcal{E}_{var} \tag{4}$$

To apply the scaling model is to use this risk as a proxy for LLM cross-entropy.

---

Because such models are mechanistic models, i.e. models a training trajectory guided by an optimizer, they can model training dynamics in new settings, e.g., data re-use (Lin et al., 2025a), or even aid in the design of algorithms, e.g., optimizers with better scaling dynamics (Ferbach et al., 2025).

Unfortunately, theoretical scaling models struggle as practical models. The naive approach of using the model's risk function to predict LLM test losses runs into challenging high-dimensional inference.

A less naive approach, which would be to use asymptotic approximations of the risk, still struggles because some of the terms are hard to approximate. Indeed, you can think of Chinchilla Approach 3 as an asymptotic approximation of the simplest terms of the theoretical models: the irreducible

error $\mathcal{E}_{irr}$, the approximation error $\mathcal{E}_{appx} = \frac{P}{N^{p-1}}$ and the bias $\mathcal{E}_{bias} = \frac{Q}{D^{p/q-1/q}}$. The variance term $\mathcal{E}_{bias}$ is the most challenging term to approximate, which may explain why no scaling model currently convincingly incorporates batch size. Lin et al. (2025b) assumed an exponentially decaying learning rate schedule, which results in a negligible variance term in the asymptotics, not suitable for analysing the effect of batch size. Paquette et al. (2025) discussed case-by-case the implication of noise under constant learning rate schedule and varying degrees of noise, with the noise becoming a bottleneck in late-stage training. However, as we will learn from the LRA elaboration to the NQS (3.2), LLM training (even at fixed lr) is not well-captured by a quadratic function optimized at fixed learning rate. The line of work by Bordelon et. al similarly did not address noise explicitly, with asymptotic results mostly concerning the training horizon and model size.

In NQS$^{++}$, we include terms to match every term of the theoretical models, including the variance term as a function of batch size. To address the challenges associated with the variance term, we take a numerical approach, rather than analytical simplifications. Crucially, the NQS can be approximated and fitted efficiently using recursions, and flexibly adapts between the phase transitions of the asymptotics.

## 3 OUR SCALING MODEL: THE NOISY QUADRATIC SYSTEM

### 3.1 DEFINITIONS

Our basic scaling model class is a close relative of theoretical scaling models with a few critical changes that make it feasible to fit to scaling data. For the sake of clarity, we introduce our model first and discuss how it relates to existing models below.

We model LLMs as infinite sequences of real numbers, and express the test loss of LLMs as a quadratic over sequences. Let $w_m^* \in \mathbb{R}^N$ be a square-summable sequence, $H : \mathbb{R}^\mathbb{N} \mapsto \mathbb{R}^\mathbb{N}$ a positive-definite linear mapping between sequences[2], and $\mathcal{E}_{\text{irr}} \geq 0$. For $w \in \mathbb{R}^\mathbb{N}$, define

$$\mathcal{Q}(w) = \mathcal{E}_{\text{irr}} + \tfrac{1}{2}\langle w - w^*, Hw - Hw^* \rangle. \tag{5}$$

$\langle w, v \rangle = \sum_m w_m v_m$ is the standard inner product. $w \in \mathbb{R}^\mathbb{N}$ represents an LLM, $w^*$ is the best LLM achievable in our model family, $\mathcal{E}_{\text{irr}}$ is the best achievable loss (the Bayes error if the model family is a universal function approximator), and $\mathcal{Q}$ is the expected test loss. Note: the coordinates $w_m$ are abstract; we don't register them with the coordinates of an LLM's weight vector.

We model LLM training as stochastic gradient descent along a finite-dimensional subspace. Let $v_n$ be an orthonormal basis of $H$'s eigenvectors, in non-increasing order of the eigenvalues $\lambda_n$. Let $\gamma, R > 0, w^{(0)} \in \mathbb{R}^\mathbb{N}, \xi_n^{(k)} \in \mathbb{R}$ be random, and $\mathbb{W}_N = \text{span}\{v_n\}_{n=1}^N$ for $N > 0$. Define the update:

$$w^{(k)} = w^{(k-1)} - \gamma \, \text{Proj}_{\mathbb{W}_N} \left( Hw^{(k-1)} - Hw^* \right) + \gamma \sum\nolimits_{n=1}^N \xi_n^{(k)} v_n. \tag{6}$$

This is an SGD optimizer of $\mathcal{Q}$ that updates $w$ along the top $N$ eigendirections of $H$ with noise injected along the same subspace. $N$ captures the model size of an LLM in our model family. Note: eigendirections don't exactly correspond to weights, but you can think of the top eigendirections as the trainable parameters of an LLM and the remaining directions as *latent*, untrained parameters.

We encode experimental observations as assumptions on $\mathcal{Q}$. Specifically, LLM test losses follow a power law in model size, which we encode with the following assumptions. Let $p > 1, P, q, Q > 0$.

(1) $\mathbb{E}[\lambda_n (\langle v_n, w^{(0)} - w^* \rangle)^2] = \frac{P}{n^p}$,     (2) $\lambda_n = \frac{Q}{n^q}$,     (3) and $\xi_n^{(k)} \sim \mathcal{N}\left(0, \lambda_n \frac{R}{B}\right)$ indep.

Assumptions (1) and (2) say: (i) for perfectly fitted models, increments in model size provide marginal improvements in the loss that diminish like a power law; (ii) for each additional increment in the model size of partially fitted models, a misestimate is discounted with a factor that decays like a separate power law. Assumption (2) is at least consistent with experimental findings that the spectra of LLM Hessians satisfy power laws (Tang et al., 2025).

Assumption (3) says: increments in model size contribute independent gradient noise that decays with the same power law as the loss discount factor ($B$ is the batch size and $R$ is a constant variance

---

[2]Technically, we also assume that $H$ is compact and self-adjoint, to invoke the spectral theorem.

factor). The independence across iterations makes it a single-epoch model. The independence across eigendirections is a strong assumption, but it is at least consistent with some experimental findings (Zhang et al., 2019) and theoretical observations (Martens, 2020).

Our scaling model class is the set of all functions that can be described as the expected value of $\mathcal{Q}$ after $K$ steps of update (6). We call this model class the *Noisy Quadratic System*. The NQS model class has at most 6 degrees of freedom; the expected value of $\mathcal{Q}$ is invariant to changes in the eigenbasis of $H$ and the step size $\gamma$ is redundant. We prove this in Appendix D. Thus, we can provide a simple expression for every element of the NQS model class, defined below.

---

**The Noisy Quadratic System of Scaling Dynamics**

**Definition 3.1. (NQS Model Class)** For integers $N, B, K > 0$, the *Noisy Quadratic System* model class consists of functions satisfying $L_\theta^{\mathrm{NQS}}(N, B, K) =$

$$\mathcal{E}_{\mathrm{irr}} + \underbrace{\sum_{n=N+1}^{\infty} \frac{P}{n^p}}_{\mathcal{E}_{\mathrm{app}}(N)} + \underbrace{\sum_{n=1}^{N} \frac{P}{n^p} \left(1 - \frac{Q}{n^q}\right)^{2K}}_{\mathcal{E}_{\mathrm{bias}}(N,K)} + \underbrace{\sum_{n=1}^{N} \sum_{k=1}^{K} \frac{RQ^2}{Bn^{2q}} \left(1 - \frac{Q}{n^q}\right)^{2K-2k}}_{\mathcal{E}_{\mathrm{var}}(N,B,K)}, \quad (7)$$

where $p > 1$, $P, q, Q, R > 0$, $\mathcal{E}_{\mathrm{irr}} \in \mathbb{R}$ and $\theta = (p, P, q, Q, R, \mathcal{E}_{\mathrm{irr}})$ are the scaling parameters.

The approximation error $\mathcal{E}_{app}$ captures the effect that latent parameters, which cannot be optimized, have on the loss. For fixed $N$, $\mathcal{E}_{\mathrm{bias}} + \mathcal{E}_{\mathrm{var}}$ captures the expected optimization error that results from imperfectly training the first $N$ dimensions: these two terms are analogous to the bias and variance in a linear regression problem, and their values depend on the number of total optimization steps $K$.

---

**Relationship with Chinchilla.** $\mathcal{E}_{\mathrm{app}}(N) \in \mathcal{O}(N^{1-p})$ decays with the same power law as Chinchilla. As we show in Appendix D, $\mathcal{E}_{\mathrm{bias}}(N, K) \in \mathcal{O}(K^{1/q - p/q})$ matches Chinchilla's second term for large values of $N, K$. The variance term, the only term that incorporates batch size $B$, doesn't have a direct analog in Chinchilla, and Chinchilla doesn't directly incorporate batch size.

**Relationship with the Noisy Quadratic Model.** Assumption (3) is derived from the covariance assumption in the Noisy Quadratic Model (NQM, Zhang et al., 2019), a model of training dynamics under rotation-invariant optimizers[3]. Yet, the NQM is not a "scaling model": the NQM doesn't model the effect of model size $N$ and it doesn't specify scaling parameters. By extending the NQM across $N$ and incorporating scaling parameters, the NQS can be fit to scaling data.

**Relationship with the Theoretical Linear Regression Models.** Assumptions (1) and (2) are derived from theoretical models of LLM scaling dynamics based on linear regression (e.g., Bahri et al., 2021; Maloney et al., 2022; Bordelon et al., 2024; Paquette et al., 2025; Lin et al., 2025b). They corresponds to the Data Generation and Modelling assumptions described in 2.2

The NQS uses a deterministic projection, removing the need to infer or marginalize out the high-dimensional random projection matrix. This allows us to approximate the variance term with numerical methods, which makes it possible to fit the NQS to scaling data. The asymptotic behavior of these quadratic models depends on the choice of assumptions, including the scaling exponents $p$ and $q$ (Paquette et al., 2025). In NQS, $p$ and $q$ are allowed to move freely between the phases between which asymptotic behaviors shift, retaining the attractive expressivity of the quadratic models.

### 3.2 Extending the Noisy Quadratic System

We found the basic NQS (def. 3.1) to be an insufficient model of LLM scaling dynamics in our experiments. We introduce two innovations to help address this, and call the fully extended model

---

[3]The NQS is a namesake of the NQM.

class, NQS$^{++}$. While we provide interpretations for these modifications, their justification is ultimately empirical. Their usefulness may depend on the LLM architecture and optimizer we used.

---

**Extensions to the Noisy Quadratic System**

**Effective Model Size (EMS).** In our experiments, compared to LLMs, the NQS displayed smaller curvature near the optimal model size. Fig. 5 Appendix E.2 contains visualizations of this failure mode. We hypothesize that the LLM weights are moving in a lower-dimensional manifold embedded within $\mathbb{R}^N$, and the number of *effective* dimensions follows its own power law in terms of the ambient weight dimension: $N_{\text{eff}}(N) = (AN)^r$, where $A, r > 0$ are additional scaling parameters of the NQS$^{++}$. We select $(A, r)$ based on the "additional variance explained" metric (26), as measured on a validation dataset, and use $\lfloor N_{\text{eff}}(N) + 1/2 \rfloor$ instead of $N$ as the first argument to the NQS.

**Learning Rate Adaptation (LRA).** The basic NQS systemically overestimated the loss for LLMs trained at small batch sizes. Given a fixed token count $D$, as one reduces $B$ and increases $K$, the NQS starts to increase, but LLM perplexity tended to maintain a flat profile [a]. Fig. 6 Appendix E.2 contains visualizations of this failure mode. The discrepancy can be corrected with step-size adaptation: at each step, LRA aims to use a step-size $\gamma$ that minimizes the expected NQS loss after the iteration, conditional on the current position. In E.6, we visualized how LRA maintains the flat profile by matching the loss trajectory of LLMs.

---
[a]In the NQM (Zhang et al. (2019)), the flat loss profile at small batch sizes resulted from co-tuning with learning rate. In our case, LLMs trained with a fixed learning rate also exhibited the flat profile.

---

Although the implementation of LRA is new, we are not the first to propose an adaptive algorithm as a theoretical explanation for LLM training behaviours. McCandlish et al. (2018) suggests that a quadratic optimization, with line search over the learning rate, is a good model of LLM loss profiles with changing batch size. Our LRA algorithm can be viewed as a crude approximation of their hypothesis. For the mathematical motivation, see Appendix D.1 of McCandlish et al. (2018).

We suspect that the normalization layers in LLMs served to regulate the norm of the weights, and therefore limited the influence of mini-batch noise, producing an effect similar to that of diminishing step-size. To reduce the computational cost of learning rate adaptation, we designed a greedy approximation scheme. This scheme incurs a small additional cost, which is linear in the number of adaptation steps. See Appendix B.3. In our experiments we found that it was important to tune the tolerance parameter of the greedy algorithm, and we recommended selecting the tolerance on a validation scaling set. Note that LRA is a deployment-time modification; we do not fit the NQS to scaling data with LRA activated.

### 3.3 LEARNING WITH THE NOISY QUADRATIC SYSTEM

For learning with the NQS and NQS$^{++}$, we adopt a model fitting and selection strategy that is highly analogous to traditional statistical models. Namely, we fit the 6 NQS scaling parameters on a train set and select models (like EMS or LRA) on validation. See Appendix B.2.

Unlike traditional learning, the design of our training and validation sets is not random. Rather than handling i.i.d. distributions, scaling models are deployed to predict critical configurations in regions with extrapolated compute budgets. So, it is particularly important that scaling models perform well at these critical points. Ideally, the model performs well for any configuration, but given that a sparsely parameterized scaling model is likely a mis-specified model, it is difficult to have high accuracy over the entire space of configurations. Thus, the modeler defines the region on which the model has to perform, potentially at the cost of deviations in other regions.

**Inferring Scaling Parameters.** For training sets, it is important to maximize coverage of configurations, but to do so strategically, as LLM training runs are expensive. We recommend training sets built from resource level sets in configuration space to balance these considerations. For this paper, we chose a scaling dataset with two components: the "IsoFLOPs" dataset and "IsoTokens" dataset. The IsoFLOPs dataset is a collection of compute $C$ level sets, each of which extends along the $N$

axis, with $B$ set at the so-called critical batch size. Similarly, the IsoTokens dataset is a collection of level sets in dataset size $D$. There are a few more details, given in Appendix F.2.

We tackle the minimization posed in equation (1) using a similar approach to Chinchilla. This minimization does not admit an analytical solution, and the loss landscape is non-convex. Chinchilla's solution is to run the BFGS algorithm locally over a range of initialization points. For NQS, we replace BFGS with a parallelize-able gradient based method. Although NQS scaling parameter gradients are relatively fast to compute using auto-differentiation, they are still slower than Chinchilla's. To address this, we parallelized over initializations. See Appendix B.2 for more details.

**Selecting Scaling Models.** Our recommendation is to design validation sets near critical points in configuration space. In our case, the validation set used medium compute budgets at least 4 times larger than the highest in training, and used LLMs runs from a small range surrounding likely optimal configurations. We select the following on validation sets: whether to use EMS or LRA, and, if so, the specific extended scaling parameters (EMS parameters and LRA tolerance).

### 3.4 COMPUTING WITH THE NOISY QUADRATIC SYSTEM

The advantages of the NQS for scaling analysis do not come at the cost of computational efficiency. Luckily, the NQS computations required can be computed efficiently via recursions, either exactly or approximately with numerical algorithms. We use the Euler-Maclaurin (EM) formulae to address the dependence on $N$ (Apostol, 1999), and the geometric series summation formula to address the dependency on $K$. Taken together, evaluations of expression (7) is $\mathcal{O}(1)$ (at most $\mathcal{O}\log(K)$ in case of numerical instability) and took about a second to compute on our hardware (including the LRA adaption procedure); fitting the NQS to the scaling dataset takes only about a minute, because we parallelize the initialization trials over multiple seeds. Details can be found in App. B.1.

## 4 EXPERIMENTS

Our experiments tested the NQS$^{++}$ model class: (i) its performance near critical points in configuration space, (ii) how its scaling predictions compared to baselines, (iii) its usefulness as a resource allocator under compound resource constraints, and (iv) its ability to select batch size schedules.

For our scaling dataset, we trained a granular (across model sizes) version of Pythia model family (for details, see Appendix F.1) with model size up to 500M. We trained models for one epoch with Adam with a fixed learning rate of $\gamma = 10^{-3}$ (Kingma & Ba, 2017). We trained on OpenWebText2 (Gokaslan & Cohen, 2019), using a customized BPE tokenizer (Gage, 1994) with a vocabulary size of 3000 and 128 sequence length. See Appendix F.2 for FLOPs budget for dataset generation.

We fit one NQS$^{++}$ model using the strategies outlined in section 3.3, and this single model is referred to as NQS$^{++}$ for all experiments below. Optimal configurations, i.e., solutions to problems

Table 2: NQS$^{++}$ outperformed Chinchilla at explaining the variance in LLM scaling dynamics near critical points in configuration space. EMS improved performance on IsoFLOP data, and LRA improved prediction on small batch sizes in IsoToken data. There was a 64x compute gap between the test runs and the most expensive train runs.

| Scaling Model | Add. train var. explained on | | Add. test var. explained on | |
|---|---|---|---|---|
| | IsoFLOPs | IsoTokens | IsoFLOPs | IsoTokens |
| Chinchilla[1] | 88 | - | −260 | - |
| NQS | 71 | −185 | 1 | 32 |
| NQS + LRA | 71 | 93 | −6 | 83 |
| NQS + EMS | **89** | −28 | 84 | 67 |
| NQS$^{++}$ | **89** | **98** | **86** | **90** |

[1] Chinchilla overfits the training data, which is also observed on its original Hoffman dataset. Please see E.1 (our data) and E.5 (Hoffman data).

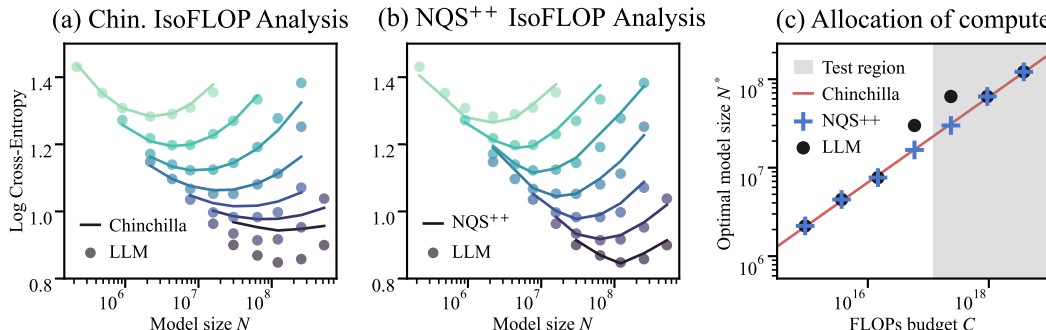

Figure 1: NQS$^{++}$ matched Chinchilla in compute allocation, and outperformed Chinchilla in predicting the loss at extrapolated compute scales. (a) and (b): for Chinchilla and NQS$^{++}$ respectively, color codes for compute budget. The 4 IsoFLOP sets from the top were used to train the scaling models. NQS$^{++}$ more accurately predicted the IsoFLOP curves at higher compute budget. (c): NQS$^{++}$ and Chinchilla performed comparably in $N/D$ allocation.

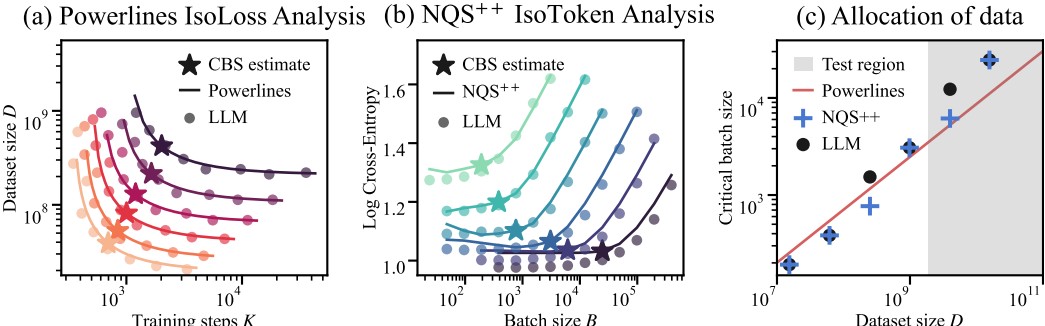

Figure 2: NQS$^{++}$ closely predicted the critical batch sizes (CBS) at out-of-sample token budgets. (a): Powerlines CBS is the batch size at the vertex of a hyperbola fitted to the IsoLoss $(K, D)$ curves. (b): NQS$^{++}$ CBS is defined as the point in $B$-LogLoss space where the IsoToken curve starts rising. (c): The differences in the definition of CBS notwithstanding, NQS$^{++}$ largely reproduced the relationship between $B_{\mathrm{crit}}$ and $D$ found in Powerlines. *Important Note: Powerlines is not expected to match LLM in (c), because the LLM points used the NQS$^{++}$ version of CBS definition.*

like eq. (3), were predicted by minimizing our fitted scaling model over a configuration grid, where $N, B$ are logrithmically spaced (at most doubling between successive values), and $K$ is computed according to the given constraints. Ground truth optima were estimated using the same grid.

Note: NQS$^{++}$ is a model of momentumless SGD in an abstract space. Nevertheless, we found it to be an acceptable model of Adam in LLM weight space. This emphasizes the point that the NQS is a mechanistic model of a process in an abstract manifold, not the domain of the weights of the LLM.

**How Well Does NQS$^{++}$ Predict LLM Test Losses?** We used a variance explained metric $\eta^2_{\mathrm{add}}$ to quantitatively evaluate scaling models. This metric compares a model's predictive performance to the best predictor given the level of compute (see Appendix C for definition).

NQS$^{++}$ outperformed Chinchilla in terms of variance explained (Table 2). On the IsoFLOP dataset, NQS$^{++}$ extrapolated well over compute scales, and maintained its predictive power on the test set, up to $\times 64$ higher in compute relative to the largest training run, and explained $86\%$ of the variance on the test set. In contrast, Chinchilla failed to estimate the loss of LLMs at out-of-sample compute budgets, potentially due to overfitting (see discussion in Appendix E.1). On the IsoTokens dataset, NQS$^{++}$ explained $90\%$ of the variation due to batch size changes, over token budgets that were up to $\times 16$ higher than the largest token budget in the training portion of the IsoTokens dataset[4].

---

[4]We did not obtain $\times 64$ on IsoTokens as this would exceed the total number of tokens in our chosen language dataset OpenWebText2.

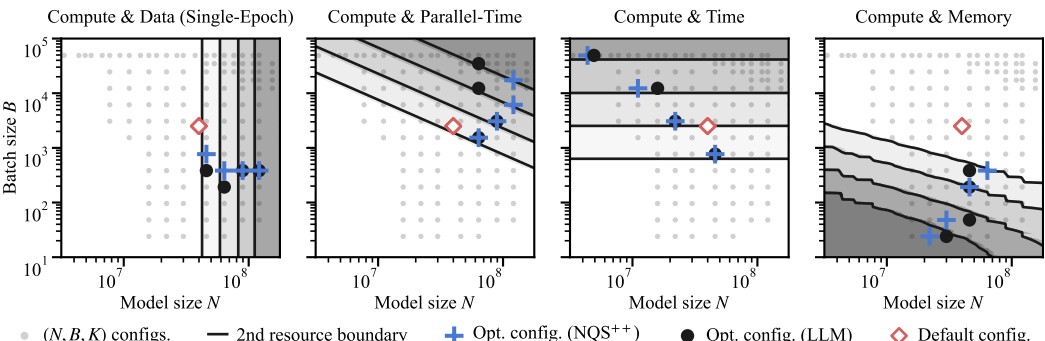

Figure 3: Under various two-resource constraints, NQS$^{++}$ selected optimal $(N, B, K)$ configurations that were close to the ground truth optimal. Each subplot: an IsoFLOP plane ($C = 2 \times 10^{17}$ FLOPs) with coordinates $(x, y)$ representing $N = x, B = y, K = {}^{2.6 \times 10^{14}}/xy$. The red diamond is the default (Chin., CBS). Four regions satisfying four progressively stricter constraints on a 2$^{\text{nd}}$ resource are shaded (darker is stricter).

**Does NQS$^{++}$ Reproduce Known Scaling Laws?** We used the NQS$^{++}$ to allocate compute and select critical batch sizes (CBSs). We compared to baselines and the ground truth to see if the NQS$^{++}$ captured known scaling law behavior. For baselines, we used Powerlines (Bergsma et al., 2025) as a method for CBS and Chinchilla (Hoffmann et al., 2022) for compute allocation. Chinchilla is trained on the training subset of the IsoFLOPs dataset, and Powerlines is trained on the training subset of the IsoTokens dataset (interpolated to obtain the IsoLoss curves).

NQS$^{++}$ and Powerlines made comparable CBS decisions, up to a slight difference in definition. Powerlines CBS $B_{\text{crit}}^{\text{PL}}(D)$ is defined as the batch size at the vertex of a hyperbola fitted to the IsoLoss $(K, D)$ curves, see Fig. 2. For NQS$^{++}$, we chose a definition of critical batch size that is more natural for the NQS$^{++}$ model family.[5] We define

$$B_{\text{crit}}^{\text{NQS}}(D; N = n) = \min \left\{ b : \frac{d}{db^2} L_{\theta^*}^{\text{NQS}}\big(N = n, B = b, K = D/(b \times \text{seq. length})\big) \geq \kappa \right\}, \quad (8)$$

where $\kappa$ is a tunable curvature threshold, and $d/db^2 L$ is approximated with finite differences using discrete values of $b$ at available data points. A prediction of $B_{\text{crit}}$ is easily obtained using NQS$^{++}$ values computed over an IsoToken set at token budget $D$ and model size $N$. NQS$^{++}$ recommended batch sizes were close to ground truth and similar to Powerlines, definition notwithstanding,

NQS$^{++}$ and Chinchilla made the same compute allocation decisions. For both, we define $N^*(C) = \operatorname{argmin}_N L_{\theta^*}^{\text{SM}}(N, D)$ subject to $6ND \leq C$. To determine a training configuration for each token budget $D$, we use $B = B_{\text{crit}}^{PL}(D)$. Both successfully found $N^*$ near the ground truth, see Fig. 1.

**NQS$^{++}$ Predicts Optimal $N, B, K$ under Compound Resource Constraints.** Compute-optimal models trained at the critical batch size are not exactly optimal (or even achievable) under some compound resource constraints. We used NQS$^{++}$ to select optimal configurations under compound constraints (defined in C), providing tailored solutions that outperformed $(N^{*\text{CHIN}}, B_{\text{crit}}^{\text{PL}})$.[6] We use two notions of time, parallel-time ($K$) and time ($NK$): with perfect model-parallelization, wall clock time is proportional to the number of iterations $K$; otherwise, $NK$ is a better indicator of training time (Bergsma et al., 2025). We also considered data constraints on $D$, in the single-epoch setting, and memory constraints on $M$, both in combination with a compute budget. NQS$^{++}$ consistently favored configurations that were nearly ground truth optimal, see Fig. 3.

---

[5]An alternative definition of critical batch size was given by Zhang et al. (2024), which required LLM evaluations along the loss gradient.

[6]Previous work (Bergsma et al., 2025) explored the $(N, B)$ efficient frontier among configurations that achieved a given loss; we address the dual problem.

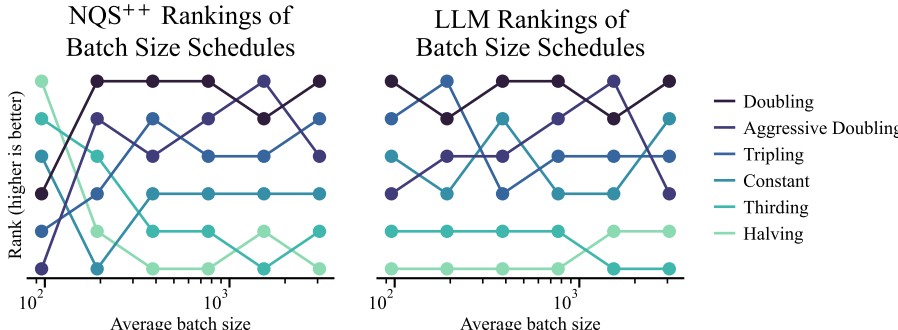

Figure 4: Batch size schedule rankings by NQS$^{++}$ were similar to LLM test loss rankings.

**What is the best way to allocate tokens through time?** A constant batch size may not be optimal. Batch size schedules are challenging to optimize for scaling heuristics because of their high-dimensionality, but NQS$^{++}$ easily incorporates schedules in the simulation of the quadratic model[7]. With a fixed number of tokens $D$, we evaluated a list of 6 different schedules, and each at 6 different *average batch size* levels. We define the average batch size to be $B_{\text{avg}} = D/K$. We found that a moderately increasing schedule was favorable over: a constant schedule, decreasing step schedules or aggressively increasing schedules.

The ranking by NQS is similar to the ground truth ranking, and the winning schedule is consistent with the choice of batch size schedule in the Llama 3 technical report (Meta AI, 2024). However, NQS$^{++}$ seems to struggle at lower average batch sizes. At these points, NQS$^{++}$ incorrectly and strongly preferred decreasing schedules. One likely culprit is the LRA in NQS$^{++}$: LRA decreases the learning rate as the batch size is decreased during training, reducing variance towards the end of training; this may not mirror how LLMs respond to drops in batch size.

## 5 CONCLUSION AND LIMITATIONS (REVISED)

We introduced the Noisy Quadratic System, a new, practical, lightweight model of LLM scaling dynamics. The NQS is designed to estimate optimal allocations of training resources whose scaling behaviour is driven by model size, batch size, and number of training steps. In our experiments, we found that the NQS allocations were close matches for the ground truth optima. We also found that the NQS predicted LLM test losses near critical training configurations very well.

**Optimizers:** To use NQS for predictions on LLMs trained on a new optimizer required re-fitting the NQS on training data with the said optimizer. In Appendix E.3, we use NQS to fit LLMs trained with SGD (rather than Adam). NQS$^{++}$ successfully fit the SGD dataset, and the the difference in the LLM optimizer was reflected in the scaling parameters: from the Adam scaling dataset, NQS$^{++}$ inferred a smaller Hessian exponent $q$, potentially reflecting Adam's pre-conditioning effect.

**Learning Rate:** Similarly, the NQS does not seem to generalize directly over learning rate. The scaling paramter $Q$ can absorb changes in $\gamma$. *A priori*, we suspected that one could increase $Q$ to predict the LLM's response to an increase in $\gamma$, but LLMs were less sensitive to changes in $\gamma$ than our quadratic system.

**Scope of Experiments:** So far, we've only tested NQS on two LLM workloads, both are Pythia-style models trained on OpenWebText2, one with SGD and the other with Adam. This limits any claims that we can make about generalization of the best scaling model across workloads. In our experiments, LLMs were trained with a constant learning rate schedule and no weight decay. We did not incorporate warm up or a cosine decay schedule. We only tested workloads at small compute scales $C < 10^{19}$ and cannot make claims about how NQS would compare to Chinchilla at larger scales.

---

[7]Previously we write $L^{\text{NQS}^{++}}$ as a function of N,B,K. In this section we update $B$ from a scalar to a step function that takes in the index set that enumerates the number of iterations $K$ and outputs the dynamic batch size. Naturally, we update the NQS$^{++}$ evaluations by scheduling the $B$ factor in the optimization of the quadratic function.

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

## A    EXTENDED RELATED WORK

**Theoretical Models of Scaling Dynamics.** The theory of scaling laws started around the early 2020s (Bahri et al., 2021; Maloney et al., 2022), where statistical models simpler than neural networks were analysed and found to exhibit similar scaling behaviors as NN. NQS$^{++}$is closely related to this family of linear regression models (Maloney et al., 2022; Paquette et al., 2025; Bordelon et al., 2024; Paquette et al., 2025). More recently, more complex models like two-layer mlps are analyzed, and found to qualitatively describe the training of NN like RNNs applied on image data (Bordelon et al., 2025; Ren et al., 2025; Arous et al., 2025). Although some of these works offer testable hypothesis (Bordelon et al., 2024; 2025), the results are limited to conjectures on the scaling exponents, and the connection with empirical results is not strong enough to warrant practical use. LLMs tends to be underexplored in the theory literature.

**The Noisy Quadratic Model and the Investigation into Critical Batch Sizes.** The pressing need to utilize the parallel computing structure initiated a line of investigation to find the best batch size that balances time efficiency and compute efficiency (Shallue et al., 2019). The Noisy Quadratic Model (NQM) (Zhang et al., 2019) was found to produce useful qualitative insights in the relationship between optimizer properties and the critical batch size. NQS$^{++}$borrows from the NQM assumptions on the noise structure of stochastic gradient updates. Inspired by similar quadratic models, quantitative scaling laws in the critical batch size are discovered (McCandlish et al., 2018), (Zhang et al., 2024),(Bergsma et al., 2025). The idea of "gradient noise scale" (McCandlish et al., 2018) is applied in the training of large scale LLMs (Brown et al., 2020).

In case where the time constraint is not severe, (Marek et al., 2025) found that smaller batch sizes are beneficial to minimizing cross-entropy under a fixed token budget; this is achievable with carefully tuned hyperparameters including those relate to the Adam optimizer.

**Scaling Laws of Learning Rate and Weight Decay.** The tuning of learning rates and weight decay are not modelled by the current version of NQS$^{++}$, but they are a key branch of scaling laws, and empirically influences the choice of batch size (Bi et al., 2024; Bjorck et al., 2025; Bergsma et al., 2025). For $lr$ selection, an alternative to scaling law is "hyperparameter transfer". Yang et al. (2022) prescribed a formula to configure neural networks, so that the optimal hyperparameters at a small scale also applied at a larger scale. Theoretical and empirical works followed to interpret and expand this regime (Dey et al., 2025; Everett et al., 2024).

**Scaling Models of Data.** Using the available data efficiently is key to scaling. NQS$^{++}$considered online training with homogeneous data, similar to (Hoffmann et al., 2022; Kaplan et al., 2020), while other works in this area explored data mixing (Shukor et al., 2025; Meta AI, 2024; Thudi et al., 2025); and training with multiple epochs (Muennighoff et al., 2025). When compared to existing practical scaling models, the NQS in its current state does not model multi-epoch training (Muennighoff et al., 2025) or data mixtures (Shukor et al., 2025), but given its close connection to theoretical works, we hope this framework can be expanded to model these configuration options and more.

**The Scaling Properties of Optimizers.** In NQS$^{++}$, we found that the optimization of a quadratic model with SGD, given the correct scaling parameters and proper elaborations, are practically sufficient to model NN trained with Adam (Kingma & Ba, 2017). Other works explicitly consider the scaling behavior of different optimizers (Zhang et al., 2019; Marek et al., 2025). Certain families of optimizers are found to outperform SGD in theory and in practice (Ferbach et al., 2025).

## B  ALGORITHMS

### B.1  COMPUTATION OF NQS AND ITS GRADIENT

This section gives details on how we efficiently compute the NQS expression (equation (7)) and its gradient with respect to the scaling parameters.

Given $(N, B, K)$ and $\theta = (P, p, Q, q, R, \mathcal{E}_{\mathrm{irr}})$, the expression we would like to evaluate is

$$L_\theta^{\mathrm{NQS}}(N, B, K) = \mathcal{E}_{\mathrm{irr}} + \underbrace{\sum_{n=N+1}^{\infty} \frac{P}{n^p}}_{\mathcal{E}_{\mathrm{app}}(N)} + \underbrace{\sum_{n=1}^{N} \frac{P}{n^p}\left(1 - \frac{Q}{n^q}\right)^{2K}}_{\mathcal{E}_{\mathrm{bias}}(N,K)} + \underbrace{\sum_{n=1}^{N}\sum_{k=1}^{K} \frac{RQ^2}{Bn^{2q}}\left(1 - \frac{Q}{n^q}\right)^{2K-2k}}_{\mathcal{E}_{\mathrm{var}}(N,K,B)}$$

$$(9)$$

$\mathcal{E}_{\mathrm{app}}(N)$ is computed using a JAX (Bradbury et al., 2018) implementation of the Riemann zeta function (in $\mathcal{O}(1)$ time).

For $\mathcal{E}_{\mathrm{bias}}(N, K)$ and $\mathcal{E}_{\mathrm{var}}(N, K, B)$:

To efficiently compute the products over $K$ and sum of products over $K$ terms, we use a divide-and-conquer algorithm that is numerically stable (Ježek, 1988). Our version is given below. This algorithm is $\mathcal{O}(\log K)$.

To efficiently compute the sums over $N$, we compute the first $5\%$ of the summation terms exactly, up till at most $N = 100$, and for the rest of the summation we approximate the sum using the corresponding integral. The integral to sum approximation is corrected with first order terms from the Euler-Maclaurin (E-M) formula. i.e. Let $L =: \min(\mathrm{int}(0.05N), 100)$, and we evaluate an expression $\sum_{n=1}^{N} f(n)$ by

$$\sum_{n=1}^{N} f(n) = \sum_{n=1}^{L} f(n) + \sum_{n=L+1}^{N} f(n)$$

$$(10)$$

---

**Algorithm 1** Calculating $S_n = \sum_{k=0}^{n-1} A^k$ and $A^n$ in $\mathcal{O}(\log n)$

---

**Require:** $A \in \mathbb{R}^{d \times d}$ and $n \geq 0$
1: **function** SUPERPOWER$(A, n)$
2:     **if** $n = 0$ **then return** $(\mathbf{0}, I)$
3:     **else**
4:         $(k, b) \leftarrow (\lfloor n/2 \rfloor, n \bmod 2)$
5:         $(S_k, A^k) \leftarrow$ SUPERPOWER$(A, k)$
6:         **if** $b = 0$ **then**
7:             **return** $(S_k + A^k S_k, A^k A^k)$
8:         **else**
9:             **return** $(S_k + A^k S_k + A^k A^k, A^k A^k A)$
10:        **end if**
11:     **end if**
12: **end function**

---

and

$$\sum_{n=L+1}^{N} f(n) \stackrel{\mathrm{E-M}}{\approx} \int_{n=L}^{N} f(n) + \frac{1}{2}(f(N) - f(L)) \tag{11}$$

Integrals are then computed with fixed 20-point Gauss-Legendre. The run time is constant in $N$.

We explicitly calculate the first few terms in the summation, because in our experiment, these terms cannot be adequately approximated with a first-order E-M formula.

To efficiently compute the gradient $\nabla_\theta L_\theta^{\mathrm{NQS}}(N, B, K)$, we first compute the gradient of the $N$-summands i.e., for $\nabla_\theta \sum_{n=L}^{N} f(n)$, we compute $\sum_{n=L}^{N} \nabla_\theta f(n)$. Since we implemented the computation of $f(n)$ in JAX (using Algorithm 1), $\nabla_\theta f(n)$ can be implemented via jax.grad$(f)$. For the summation over $N$, analogously, we evaluate the first few terms exactly, and then approximate the rest with an integral.

$$\sum_{n=1}^{N} \nabla_\theta f(n) \approx \sum_{n=1}^{L} \nabla_\theta f(n) + \int_{n=L}^{N} \nabla_\theta f(n) + \frac{1}{2}(\nabla_\theta f(N) - \nabla_\theta f(L)) \tag{12}$$

The computations are implemented with JAX and parallelize-able, making it possible to fit the scaling model efficiently, by parallelizing over random initialization trials.

### B.2 FITTING NQS TO SCALING DATA

First, we describe how to fit an NQS system on the training data, assuming the hyper-parameters (for the extensions) are determined. Then we describe how to select these hyper-parameters using a validation dataset.

#### B.2.1 INFERENCE

Given a scaling dataset $\left\{ (N_i, B_i, K_i), L_i^{\mathrm{NN}} \right\}_{i=1}^{m}$, the goal of fitting an NQS is to find $\theta$ that minimizes the scaling loss given by (1):

$$\theta^* = \operatorname*{argmin}_\theta \frac{1}{|\mathrm{train}|} \sum_{i \in \mathrm{train}} \mathcal{L}(L_\theta^{\mathrm{SM}}(N_i, B_i, K_i), L_i). \tag{13}$$

In our experiments, we took $\mathcal{L}$ to be the Huber loss between the logarithms of its two arguments, as in Hoffmann et al. (2022).

**Data Filtering.** As described in section 3.3, the training portion of the scaling dataset is composed of the IsoFlops training dataset and the IsoTokens training dataset. Not all elements of the IsoTokens

training dataset are suitable to be included in the scaling loss. Recall that LRA is a deployment time modification. Because we do not have an implementation of $\nabla_\theta(L^{\text{NQS}})$ that incorporates LRA, we would like to remove training data points that are expected to be significantly affected by LRA. In our observations, it suffices to remove data points with $(N, B, K)$ satisfying the following: $L^{\text{NN}}(N, B/2, 2K) > L^{\text{NN}}(N, B, K) - 0.05$. We have access to this information because in the IsoTokens dataset, $B$ are spaced logarithmically, where the successive points are doubled in $B$. This is a rule of thumb that has resulted in a good fit on the filtered training dataset.

**Optimization.** Over the filtered portion of the training dataset, we optimized the target loss in Eq. (1) using the Adam optimizatior, over parallelized random initialization trials, using gradients estimated according to Appendix B.1. Details are given below:

- Initialisations: we used 1000 pseudo-random initialisations, spaced as a Latin hyper-cube over the following range: $p \in [1.05, 2.5], P \in [0.5, 100], q \in [0.6, 2.5], Q \in [0.05, 20], \sqrt{R} \in [0.1, 10], \mathcal{E}_{\text{irr}} \in [0.1, 1.5]$. Note that these values are allowed to move outside of these ranges during the optimization. In the implementation, we parametrized $R$ with $\sqrt{R}^2$.

- Optimization: we used the standard Adam optimizer with gradient clipping (gradients clipped to be within $[-1.0, 1.0]$). Each optimization trial lasts for 1000 iterations.

- Decision: we picked the lowest loss iteration for each random initialization, and then compared them across the initializations to select the final scaling parameters.

In our experiments, the optimization process takes about 1-2 hours (on one H100 GPU).

### B.2.2 Hyperparameter Selection for NQS$^{++}$

**Power law scaling parameters for EMS.** We describe one procedure to select EMS hyperparameters $(A, r)$. Recall that $N_{\text{eff}}(N) = (AN)^r$.

1. Fix $A = 1$, among $[0.55, 0.6, 0.75, 0.9, 1.0]$, select a ratio $r$ such that a scaling model trained with hyper-parameters $A, r$ maximizes the additional variance explained metric in the validation set. (denote $r_1$).

2. Fix $r = 1$, among $[0.001, 0.01, 0.1, 1]$, select a multiplier $A$ that maximize the additional variance explained metric. (denote $A_2$).

3. Select 5 points, approximately evenly spaced along the line segment between $(1, r_1)$ and $(A_2, 1)$, using log scale for $A$ and normal scale for $r$. Test these points and select the one with the maximum additional variance explained metric.

**Tolerance for LRA.** In B.3 we go into details of the LRA algorithm. In short, the LRA is a greedy algorithm that decays learning rate at certain steps during the optimization of the quadratic system, where the decay results in an improvement in the expected value of the quadratic function. We place a tolerance on the minimum amount of improvement before a learning rate decay is triggered.

Since LRA is a deployment time modification, tuning the tolerance parameter does not require re-fitting of the system. It is recommended to determine the EMS paramters first, then use a validation set to determine the appropriate tolerance (note: in case where an IsoTokens validation set is not available, an IsoFlops validation set would also suffice for this task).

### B.3 Learning Rate Adaptation

In LRA, we search for a step-function *learning rate schedule* of length $K$ that improves the expected loss of the quadratic $\mathbb{E}\left[\mathcal{Q}_\theta(w^{(K)})\right]$, and then outputs the expected loss with said schedule. By learning rate schedule, we mean a sequence: $k \mapsto \gamma_k$, where $\gamma_k$ is the learning rate used in the $k^{th}$ update of $w$. For this algorithm, we restrict the learning rate schedule to be a step function, with evenly spaced steps. Details of the algorithm is given in Algorithm 2. We denote by $L(\text{lr\_sch\_curr})$ the expected loss of the quadratic optimized with a learning rate schedule (a sequence) of lr\_sch\_curr. The length of the learning rate schedule dictates the number of steps that the quadratic function is optimized for.

---

**Algorithm 2** Learning Rate Adaptation

---

1: **Input:** Loss function $L(\cdot)$, total steps $K$, number of stages $S$, threshold
2: **Output:** Optimized learning rate schedule and corresponding loss
3: Compute step lengths: $h_s = \lfloor K/S \rfloor$ for $s < S$, and $h_S = K \bmod S$
4: Initialize learning rate schedule (lr_sch) as a sequence of 1's of length $h_1$.
5: prev_stage_lr $\leftarrow$ lr_sch$[-1]$
6: **for** $s = 2$ to $S$ **do**
7:     **if** $h_s = 0$ **then**
8:         **break**
9:     **end if**
10:     lr_sch_curr $\leftarrow$ lr_sch.append(repeat(prev_stage_lr, $h_s$))
11:     $L_{\text{curr}} \leftarrow L(\text{lr\_sch\_curr})$
12:     prev_attempt_lr $\leftarrow$ lr_sch_curr$[-1]$
13:     lr_sch_new $\leftarrow$ lr_sch.append(repeat(prev_attempt_lr $\times$ 0.5, $h_s$))
14:     $L_{\text{new}} \leftarrow L(\text{lr\_sch\_new})$
15:     **while** $L_{\text{new}} - L_{\text{curr}} < -\text{threshold}$ **do**
16:         $L_{\text{curr}} \leftarrow L_{\text{new}}$
17:         lr_sch_curr $\leftarrow$ lr_sch_new
18:         prev_attempt_lr $\leftarrow$ lr_sch_curr$[-1]$
19:         lr_sch_new $\leftarrow$ lr_sch.append(repeat(prev_attempt_lr $\times$ 0.5, $h_s$))
20:         $L_{\text{new}} \leftarrow L(\text{lr\_sch\_new})$
21:     **end while**
22:     lr_sch $\leftarrow$ lr_sch_curr
23:     prev_stage_lr $\leftarrow$ lr_sch$[-1]$
24: **end for**
25: **return** lr_sch_curr, $L_{\text{curr}}$

---

An input to the Algorithm is tolerance: this value controls the "greediness" of the weight decay, and only an improvement beyond the tolerance can trigger a decay in the learning rate. This value should be tuned using a validation scaling dataset (see Section B.2.2).

The algorithm as given is $\mathcal{O}(S^2 \log K)$ in run time, where $S$ is the maximum number of change points allowed in the learning rate schedule. The dependence on $S$ is quadratic, because computing $L(\text{lr sch})$ from scratch takes $\mathcal{O}(S \log K)$ time. However, by carefully caching the relevant values from the computation of $L(\text{lr sch curr})$, one can compute $L(\text{lr sch new})$ in $\mathcal{O}(\log K)$ time.

To understand this, let us start by looking at the variance term of $L$ for a single dimension, say the $n^{th}$ eigen direction of the Hessian matrix of the quadratic. Assume we have a 3-stage learning rate schedule. The stages are $A, B, C$, with learning rates $[\gamma_A, \gamma_B, \gamma_C]$. Each stage lasts for $T$ weight updates. The variance in dimension $n$ is

$$\mathcal{E}_{\text{var},n} =: \frac{1}{2} \sum_{k=1}^{3T} \gamma_k^2 \frac{\lambda_n R}{B} \prod_{j=k}^{3T} (1 - \gamma_j \lambda_n)^2, \tag{14}$$

where $\lambda_n = \frac{Q}{n^q}$ is the $n^{th}$ eigenvalue of the operator $H$. (We derived the expression for $\mathcal{E}_{\text{var}}$ in D for constant learning rate, which is easily extended to a step schedule.) The term that depends on $K$

(and thus $S$) is:

$$\mathcal{E}_{\text{var},n}/(\frac{\lambda_n R}{2B}) = \sum_{k=1}^{3T} \prod_{j=k}^{3T} \gamma_k^2 (1 - \gamma_j \lambda_n)^2 \tag{15}$$

$$= \sum_{k=1}^{T} \prod_{j=k}^{3T} \gamma_k^2 (1 - \gamma_j \lambda_n)^2 + \sum_{k=T+1}^{2T} \prod_{j=k}^{3T} \gamma_k^2 (1 - \gamma_j \lambda_n)^2 + \sum_{k=2T+1}^{3T} \prod_{j=k}^{3T} \gamma_k^2 (1 - \gamma_j \lambda_n)^2 \tag{16}$$

$$= \sum_{k=1}^{T} \gamma_k^2 \prod_{j=k}^{T} (1 - \gamma_j \lambda_n)^2 \prod_{j=T+1}^{2T} (1 - \gamma_j \lambda_n)^2 \prod_{j=2T+1}^{3T} (1 - \gamma_j \lambda_n)^2 + ... + ... \tag{17}$$

$$= \sum_{k=1}^{T} \gamma_A^2 \prod_{j=k}^{T} (1 - \gamma_A \lambda_n)^2 \prod_{j=T+1}^{2T} (1 - \gamma_B \lambda_n)^2 \prod_{j=2T+1}^{3T} (1 - \gamma_C \lambda_n)^2 + ... + ... \tag{18}$$

$$= (1 - \gamma_B \lambda_n)^{2T} (1 - \gamma_C \lambda_n)^{2T} \sum_{k=1}^{T} \gamma_A^2 (1 - \gamma_A \lambda_n)^{2(T-k)} + ... + ... \tag{19}$$

$$\tag{20}$$

Define $F_n(\gamma) = (1 - \gamma \lambda_n)^{2T}$ and $G_n(\gamma) = \sum_{k=1}^{T} \gamma^2 (1 - \gamma \lambda_n)^{2(T-k)}$. We can now write a recursion in the stages :

$$\mathcal{E}_{\text{var},n}/(\frac{\lambda_n R}{2B}) \text{ at stage } C = G_A(\gamma_A) F_B(\gamma_B) F_C(\gamma_C) + G_B(\gamma_B) F_C(\gamma_C) + G_C(\gamma_C) \tag{21}$$

$$= \Big( G_A(\gamma_A) F_B(\gamma_B) + G_B(\gamma_B) \Big) F_C(\gamma_C) + G_C(\gamma_C) \tag{22}$$

$$= \Big\{ \mathcal{E}_{\text{var},n}/(\frac{\lambda_n R}{2B}) \text{ at stage } B \Big\} \times F_C(\gamma_C) + G_C(\gamma_C) \tag{23}$$

Similarly, we can write the bias term as a recursion:

$$\mathcal{E}_{\text{bias, n}}/(\frac{P}{2n^p}) \text{ at stage C} = \prod_{k=1}^{3T} (1 - \gamma_k \lambda_n)^2 = F_A(\gamma_A) F_B(\gamma_B) F_C(\gamma_C) \tag{24}$$

$$= \Big\{ \mathcal{E}_{\text{bias, n}}/(\frac{P}{2n^p}) \text{ at stage B} \Big\} \times F_C(\gamma_C) \tag{25}$$

To go from $N = n$ to the full risk, we need to sum the above expressions over $n = 1, ..., N$. As described previously, we estimate the sum over $N$ with a fixed-point Gaussian quadrature. Instead of computing the expression at $\mathcal{E}_{\text{bias},n}, \mathcal{E}_{\text{var},n}$, we can compute $\mathcal{E}_{\text{bias},m}, \mathcal{E}_{\text{var},m}$ at 20 values of $m$ spaced between 1 and $N$. The rest is straightforward.

## C  DEFINITIONS

**Additional Variance Explained.**  On a scaling dataset, $\eta_{\text{add}}^2$ is defined as:

$$\eta_{\text{add}}^2 = 1 - \frac{\sum_{c \in C} \sum_{i \in S_c} \Big( \log L_i^{\text{LLM}} - \log L_i^{\text{NQS}}(N_i, B_i, K_i) \Big)^2}{\sum_{c \in C} \sum_{i \in S_c} \Big( \log L_i^{\text{LLM}} - \sum_{i \in S_c} \log L_i^{\text{LLM}} \Big/ |S_c| \Big)^2}, \tag{26}$$

where $c \in C$ are compute budgets within the scaling dataset ($C = \{1e15, ..., 4e18\}$), and $S_c = \{i : 6N_i B_i K_i = c\}$ is the set of all data points at the compute level $c$.

**Doubly Constrained Optimal Configurations.**  For a doubly constrained setup, we define the constrained optimal configuration as:

$$(N, B, K)^*(f, c) = \operatorname*{argmin}_{(N,B,K)} L(N, B, K) \; s.t. \; F \leq f, C \leq c \text{ for } F \in \{D, N, NK, M\}.$$

To obtain the NQS$^{++}$ prediction of the optima, we ran NQS$^{++}$ predictions along a grid over $(N, B, K)$ in the IsoFlop plane where $C(N, B, K) = c$, and selected the configuration with the lowest predicted loss.

# D PROOFS

## D.1 DEGREES OF FREEDOM OF THE NQS

Before the derivation, let us review the assumptions and requirements in section 3.

We model LLMs as infinite sequences of real numbers, and express the test loss of LLMs as a quadratic over sequences. Let $w_m^* \in \mathbb{R}$ be an square-summable sequence, $H : \mathbb{R}^{\mathbb{N}} \mapsto \mathbb{R}^{\mathbb{N}}$ a positive-definite linear mapping between sequences[8], and $\mathcal{E}_{\mathrm{irr}} \geq 0$. For $w \in \mathbb{R}^{\mathbb{N}}$, define

$$\mathcal{Q}(w) = \mathcal{E}_{\mathrm{irr}} + \tfrac{1}{2}\langle w - w^*, Hw - Hw^* \rangle. \tag{27}$$

We model LLM training as stochastic gradient descent along an finite-dimensional subspace. Let $v_n$ be an orthonormal basis of $H$'s eigenvectors, in non-increasing order of the eigenvalues $\lambda_n$. Let $\gamma, R > 0, w^{(0)} \in \mathbb{R}^{\mathbb{N}}, \xi_n^{(k)} \in \mathbb{R}$ be random, and $\mathbb{W}_N = \mathrm{span}\{v_n\}_{n=1}^N$ for $N > 0$. Define the update:

$$w^{(k)} = w^{(k-1)} - \gamma \, \mathrm{Proj}_{\mathbb{W}_N} \left( Hw^{(k-1)} - Hw^* \right) + \gamma \sum\nolimits_{n=1}^N \xi_n^{(k)} v_n. \tag{28}$$

We model this with the following assumptions. Let $p > 1, P, q, Q > 0$.

(1) $\mathbb{E}[\lambda_n \times (\langle v_n, w^{(0)} - w^* \rangle)^2] = P/n^p$,

(2) $\lambda_n = Q/n^q$,

(3) and $\xi_n^{(k)} \sim \mathcal{N}(0, \sqrt{\lambda_n \times (R/B)})$ independently.

We want to show that $\mathbb{E}[\mathcal{Q}(w^{(K)})] =$

$$\mathcal{E}_{\mathrm{irr}} + \underbrace{\sum_{n=N+1}^{\infty} \frac{P}{n^p}}_{\mathcal{E}_{\mathrm{app}}(N)} + \underbrace{\sum_{n=1}^{N} \frac{P}{n^p} \left( 1 - \frac{Q}{n^q} \right)^{2K}}_{\mathcal{E}_{\mathrm{bias}}(N,K)} + \underbrace{\sum_{n=1}^{N} \sum_{k=1}^{K} \frac{RQ^2}{Bn^{2q}} \left( 1 - \frac{Q}{n^q} \right)^{2K-2k}}_{\mathcal{E}_{\mathrm{var}}(N,K,B)} \tag{29}$$

which is the expression we use for the NQS model family. We would also show that the NQS model family, defined as $L^{\mathrm{NQS}}(N, B, K) = \mathbb{E}[\mathcal{Q}(w^{(K)})]$, has at most 6 degrees of freedom.

**Proof.** The update rule gives

$$w^{(k)} - w^{(k-1)} = -\gamma \, \mathrm{Proj}_{\mathbb{W}_N} \left( H(w^{(k-1)} - w^*) \right) + \gamma \sum\nolimits_{n=1}^N \xi_n^{(k)} v_n. \tag{30}$$

$$= -\gamma \, \mathrm{Proj}_{\mathbb{W}_N} \left( H \sum_{n=1}^{\infty} \left\langle (w^{(k-1)} - w^*), v_n \right\rangle v_n \right) + \gamma \sum\nolimits_{n=1}^N \xi_n^{(k)} v_n. \tag{31}$$

$$= -\gamma \, \mathrm{Proj}_{\mathbb{W}_N} \left( \sum_{n=1}^{\infty} \left\langle (w^{(k-1)} - w^*), v_n \right\rangle \lambda_n v_n \right) + \gamma \sum\nolimits_{n=1}^N \xi_n^{(k)} v_n. \tag{32}$$

$$= -\gamma \sum_{n=1}^{N} \left\langle (w^{(k-1)} - w^*), v_n \right\rangle \lambda_n v_n + \gamma \sum\nolimits_{n=1}^N \xi_n^{(k)} v_n. \tag{33}$$

For each $n \leq N$,

$$\left\langle w^{(k)} - w^{(k-1)}, v_n \right\rangle = -\gamma \left\langle (w^{(k-1)} - w^*), v_n \right\rangle \lambda_n + \gamma \xi_n^{(k)} \tag{34}$$

$$\left\langle w^{(k)} - w^*, v_n \right\rangle = \left\langle w^{(k)} - w^{(k-1)}, v_n \right\rangle + \left\langle w^{(k-1)} - w^*, v_n \right\rangle = (1 - \gamma \lambda_n) \left\langle (w^{(k-1)} - w^*), v_n \right\rangle + \gamma \xi_n^{(k)}. \tag{35}$$

Thus $\mathbb{E}\left[ \left( \left\langle w^{(k)} - w^*, v_n \right\rangle \right)^2 \right]$

$$= (1 - \gamma \lambda_n)^2 \mathbb{E}\left[ (\left\langle (w^{(k-1)} - w^*), v_n \right\rangle)^2 \right] + \gamma^2 \mathbb{E}\left[ (\xi_n^{(k)})^2 \right] \tag{36}$$

---

[8]Technically, we also assume that $H$ is compact and self-adjoint, to invoke the spectral theorem.

Apply recursively, we get $\mathbb{E}\left[\left(\langle w^{(k)} - w^*, \; v_n \rangle\right)^2\right]$

$$= (1 - \gamma\lambda_n)^{2k}\mathbb{E}\left[\left(\langle (w^{(0)} - w^*), \; v_n \rangle\right)^2\right] + \sum_{j=1}^{k}(1 - \gamma\lambda_n)^{2(k-j)}\gamma^2\mathbb{E}\left[(\xi_n^{(j)})^2\right] \tag{37}$$

$$= (1 - \gamma\lambda_n)^{2k}\frac{1}{\lambda_n}\frac{P}{n^p} + \gamma^2\sum_{j=1}^{k}(1 - \gamma\lambda_n)^{2(k-j)}\lambda_n\frac{R}{B} \tag{38}$$

We also know $w^{(k)} - w^{(0)} \in \mathrm{span}\{v_1, ...v_N\}$, so $\langle w^{(k)} - w^{(0)}, \; v_n \rangle = 0$ for any $n > N$.

$$\mathbb{E}\left[\left\langle w^{(k)} - w^*, \; H(w^{(k)} - w^*)\right\rangle\right] \tag{39}$$

$$= \mathbb{E}\left[\sum_{n=1}^{N}\lambda_n\left\langle w^{(k)} - w^{(0)}, \; v_n \right\rangle^2 + \sum_{n=1}^{N}\lambda_n 2\left\langle w^{(k)} - w^{(0)}, \; v_n \right\rangle\left\langle w^{(0)} - w^*, \; v_n \right\rangle + \sum_{n=1}^{\infty}\lambda_n\left\langle w^{(0)} - w^*, \; v_n \right\rangle^2\right] \tag{40}$$

$$= \sum_{n=1}^{N}\lambda_n\mathbb{E}\left[\left\langle w^{(k)} - w^{(0)}, \; v_n \right\rangle^2\right] + \sum_{n=N+1}^{\infty}\mathbb{E}\left[\lambda_n(\left\langle w^{(0)} - w^*, \; v_n \right\rangle)^2\right] \tag{41}$$

$$= \sum_{n=1}^{N}\lambda_n(1 - \gamma\lambda_n)^{2k}\frac{1}{\lambda_n}\frac{P}{n^p} + \sum_{n=1}^{N}\lambda_n\gamma^2\sum_{j=1}^{k}(1 - \gamma\lambda_n)^{2(k-j)}\lambda_n\frac{R}{B} + \sum_{n=N+1}^{\infty}\frac{P}{n^p}. \tag{42}$$

Therefore $\mathbb{E}[\mathcal{Q}(w^{(K)})] = \mathcal{E}_{\mathrm{irr}} + \frac{1}{2}\mathbb{E}\left[\left\langle w^{(K)} - w^*, \; H(w^{(K)} - w^*)\right\rangle\right]$

$$= \mathcal{E}_{\mathrm{irr}} + \frac{1}{2}\sum_{n=1}^{N}(1 - \gamma\lambda_n)^{2K}\frac{P}{n^p} + \frac{1}{2}\sum_{n=1}^{N}\lambda_n^2\gamma^2\sum_{k=1}^{K}(1 - \gamma\lambda_n)^{2(K-k)}\frac{R}{B} + \frac{1}{2}\sum_{n=N+1}^{\infty}\frac{P}{n^p} \tag{43}$$

$$= \mathcal{E}_{\mathrm{irr}} + \frac{1}{2}\sum_{n=1}^{N}(1 - \gamma\frac{Q}{n^q})^{2K}\frac{P}{n^p} + \frac{1}{2}\sum_{n=1}^{N}\frac{Q^2}{n^{2q}}\frac{R}{B}\gamma^2\sum_{k=1}^{K}(1 - \gamma\frac{Q}{n^q})^{2(K-k)} + \frac{1}{2}\sum_{n=N+1}^{\infty}\frac{P}{n^p}. \tag{44}$$

By re-parameterizing $Q =: \gamma Q, R =: R/2, P =: P/2$, we get:

$$\mathbb{E}[\mathcal{Q}(w^{(K)})] \tag{45}$$

$$= \mathcal{E}_{\mathrm{irr}} + \sum_{n=1}^{N}(1 - \frac{Q}{n^q})^{2K}\frac{P}{n^p} + \sum_{n=1}^{N}\frac{Q^2}{n^{2q}}\frac{R}{B}\sum_{k=1}^{K}(1 - \frac{Q}{n^q})^{2(K-k)} + \sum_{n=N+1}^{\infty}\frac{P}{n^p}. \tag{46}$$

Other than $N, B, K$, this function has 6 input arguments: $P, p, Q, q, R$ and $\mathcal{E}_{\mathrm{irr}}$. Thus, the model class $L^{\mathrm{NQS}}(N, B, K) = \mathbb{E}[\mathcal{Q}(w^{(K)})]$ has at most 6 degrees of freedom.

**End of proof.**

## D.2 Asymptotic Upper Bound for the Bias Term

In this section we show that $\mathcal{E}_{\text{bias}}(N, K) = \frac{1}{2} \sum_{n=1}^{N} (1 - \gamma \frac{Q}{n^q})^{2K} \frac{P}{n^p}$ is $\mathcal{O}(K^{-(p/q-1/q)})$.

**Proof.**

$$\mathcal{E}_{\text{bias}}(N, K) = \frac{1}{2} \sum_{n=1}^{N} (1 - \gamma \frac{Q}{n^q})^{2K} \frac{P}{n^p} \tag{47}$$

$$\leq \frac{P}{2} \sum_{n=1}^{N} n^{-p} \prod_{k=1}^{K} \exp(-\gamma Q n^{-q})^2 \tag{48}$$

$$= \frac{P}{2} \sum_{n=1}^{N} n^{-p} \exp(-2K\gamma Q n^{-q}) \tag{49}$$

We next bound the summation with integrals. To do that, we need to find the regions where the summand is monotone. Take the derivative of the summand $f(n) = n^{-p} \exp(-2K\gamma Q n^{-q})$:

$$\frac{d}{dn} f(n) = (-p)n^{-p-1} \exp(-2K\gamma Q n^{-q}) + n^{-p} \exp(-2K\gamma Q n^{-q})(-2K\gamma Q)(-q)n^{-q-1} \tag{50}$$

$$= pn^{-p-1} \exp(-2K\gamma Q n^{-q}) \left( \frac{2q\gamma Q}{p} \frac{K}{n^q} - 1 \right) \tag{51}$$

Define $h(K) = (\frac{2q\gamma QK}{p})^{1/q}$. The summand is non-decreasing in $n$ for $1 \leq n \leq h(K)$, and non-increasing for $h(K) \leq n \leq N$. Using this monotonicity:

$$\mathcal{E}_{\text{bias}}(N, K) = \frac{P}{2} \sum_{n=1}^{\lfloor h(K) \rfloor} f(n) + \sum_{\lceil h(K) \rceil}^{N} f(n) \tag{52}$$

$$\leq \frac{P}{2} \int_{n=1}^{\lfloor h(K) \rfloor + 1} f(n)dn + \int_{\lceil h(K) \rceil - 1}^{N} f(n)dn \tag{53}$$

$$\leq \frac{P}{2} \int_{n=1}^{\lfloor h(K) \rfloor} f(n)dn + 2f(h(K)) + \int_{\lceil h(K) \rceil}^{N} f(n)dn \tag{54}$$

$$\leq \frac{P}{2} \left( \int_{1.5}^{\lfloor h(K) \rfloor + 0.5} f(n)dn + 2f(h(K)) + \int_{\lceil h(K) \rceil - 0.5}^{N - 0.5} f(n)dn \right) \tag{55}$$

Simplify the integral

$$\int_{x_1}^{x_2} f(x)dx = \int_{x_1}^{x_2} x^{-p} \exp(-cKx^{-q})dx \tag{56}$$

$$= \int_{t_1 = cKx_1^{-q}}^{t_2 = cKx_2^{-q}} (cK/t)^{-p/q} \exp(-t) \frac{d(cK/t)^{1/q}}{dt} dt \tag{57}$$

$$= \int_{t_1 = cKx_1^{-q}}^{t_2 = cKx_1^{-q}} (cK/t)^{-p/q} \exp(-t)(cK)^{1/q}(-1/q)t^{-1/q-1}dt \tag{58}$$

$$= (1/q)(cK)^{-(p/q-1/q)} \int_{t_2 = cKx_2^{-q}}^{t_1 = cKx_1^{-q}} \exp(-t)t^{p/q-1/q-1}dt \tag{59}$$

Define $G(s, (t_1, t_2)) = \int_{t_1}^{t_2} t^{s-1} \exp(-t)dt$ and $c = 2\gamma Q$.

Then we have

$$\mathcal{E}_{\text{bias}}(N, K) \leq \frac{P}{2}\frac{1}{b}(cK)^{-(p/q-1/q)}\Bigg( \tag{60}$$

$$G(p/q - 1/q, \, (cK(\lfloor h(K)\rfloor + 0.5)^{-q}, \, cK(1.5)^{-q})) + \tag{61}$$

$$+ 2f(h(K)) + \tag{62}$$

$$G(p/q - 1/q, \, (cK(N - 0.5)^{-q}, \, cK(\lceil h(K)\rceil - 0.5)^{-q}))\Bigg) \tag{63}$$

for convenience, if $y$ is an integer, define $\lfloor y \rfloor = y$ and $\lceil y \rceil = y + 1$, so that we always have $\lfloor y \rfloor + 0.5 = \lceil y \rceil - 0.5$.

Then we get $\frac{\mathcal{E}_{\text{bias}}(N,K)}{\frac{P}{2}(cK)^{-(p/q-1/q)}} \leq 2f(h(K)) +$

$$G(p/q - 1/q, \, (cK(N - 0.5)^{-q}, \, cK(1.5)^{-q})) \tag{64}$$

$$\leq 2f(h(K)) + G\Big(p/q - 1/q, \, (0, \, \infty)\Big) \tag{65}$$

$$\leq 2f(h(K)) + \Gamma(p/q - 1/q) \tag{66}$$

$$\mathcal{E}_{\text{bias}}(N, K) \leq \frac{P}{2}(\frac{1}{2\gamma Q})^{p/q-1/q}\Big(2f(h(K)) + \Gamma(p/q - 1/q)\Big) K^{-(p/q-1/q)} \tag{67}$$

$$f(h(K)) \propto K^{-p/q} \to 0 \text{ as } K \to \infty. \tag{68}$$

We can find sufficiently large $M_1$ such that for all $K > M_1$, $f(h(K)) \leq$ e.g. $\Gamma(p/q - 1/q)$ (or any other constant). Therefore $\mathcal{E}_{\text{bias}}(N, K)$ is $\mathcal{O}(K^{-(p/q-1/q)})$. (Holds for any $N$ sufficiently large.)

**End of Proof.**

# E    FIGURES AND TABLES

With the exception of figures 8 and 7, the figures and tables in this section are based on NQS fitted to LLMs trained with the Adam optimizer.

## E.1    COMPARISONS WITH CHINCHILLA

In Table 2, we saw that NQS$^{++}$was predictive with a $\times 64$ compute gap, and the test performance (86%) is comparable to that on training (89%). In contrast, Chinchilla fitted the training dataset very well (88%), but failed to predict the loss of LLMs in the test set (-260%). Upon investigation, the error on the test set was mostly due to Chinchilla overestimating the overall level of LLM test loss at the test compute budgets. In Table 3, as we close the compute gap between train and test, Chinchilla's test metric improved, and training metric deteriorated. Chinchilla seemed to have overfitted on our scaling dataset.

Table 3: In our experiments, Chinchilla overfitted on small datasets. As more data is added, Chinchilla's performance on training deteriorated, and performance on test improved.

| | Add. var. explained | | Compute gap |
| --- | --- | --- | --- |
| Chinchilla fitted on | Train | Test | |
| Train | 88 | −260 | up to 64x |
| Train + val. | 87 | −113 | up to 16x |
| Train + val. + part of test | 82 | 27 | 4x |
| Train + val. + all of test | 81 | 52 | None |

## E.2 ABLATION STUDIES

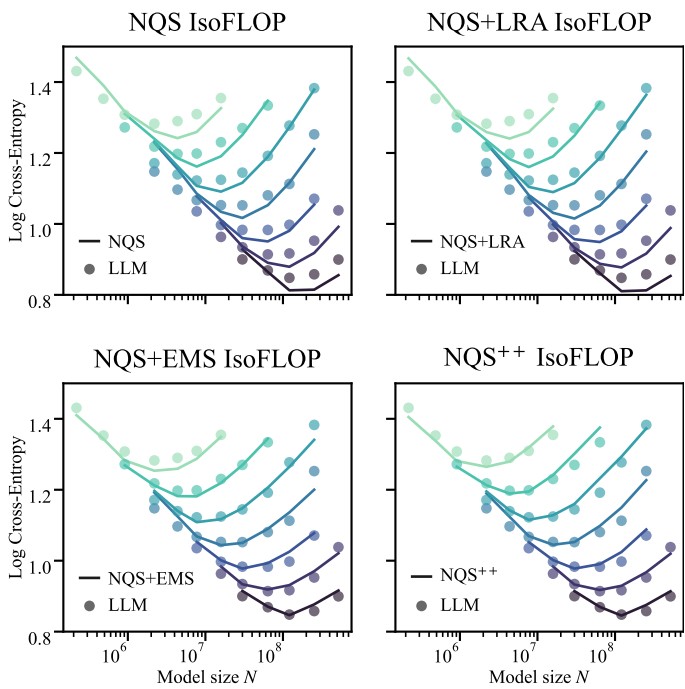

Figure 5: NQS without EMS fits IsoFLOPs poorly.

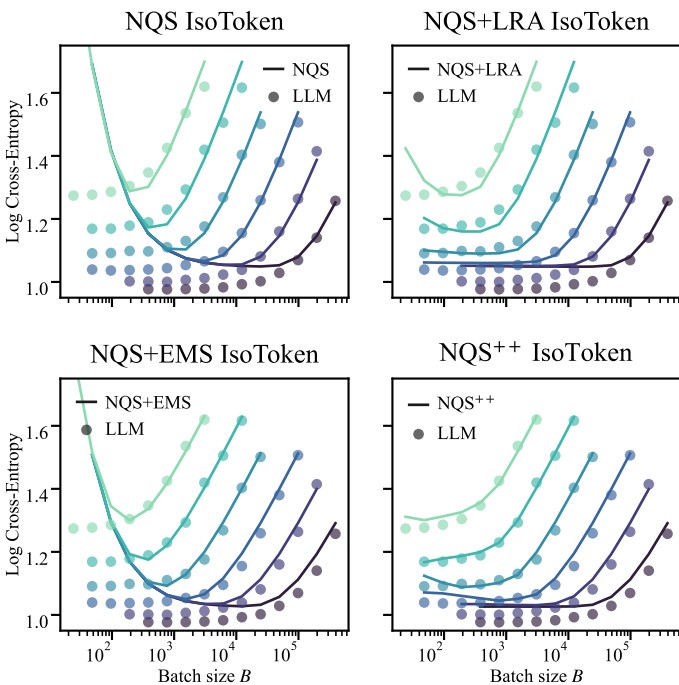

Figure 6: NQS needs both EMS and LRA to fit IsoTokens well, but the LRA accounts for most of the improvements.

## E.3 FITTING NQS TO LLMS TRAINED WITH SGD

Table 4: On LLMs trained with SGD, NQS$^{++}$outperformed Chinchilla on extrapolated compute budgets (IsoFlops), and explained 80% of the variance due to variation in batch sizes (IsoTokens). Note that on the IsoFLOPs test set, both Chinchilla and NQS$^{++}$gave negative variance-explained values: this was due to the flatness of the IsoFLOP curves in the test set; the variance within each FLOPS budget was smaller than the squared difference between the LLM loss and the Scaling Model loss. The average squared difference between NQS$^{++}$and LLM is small, as visible in Fig. 7.

| | Add. train var. explained on | | Add. test var. explained on | |
|---|---|---|---|---|
| Scaling Model | IsoFLOPs | IsoTokens | IsoFLOPs | IsoTokens |
| Chinchilla | **98** | - | −1960 | - |
| NQS$^{++}$ | 89 | **97** | **−58** | **80** |

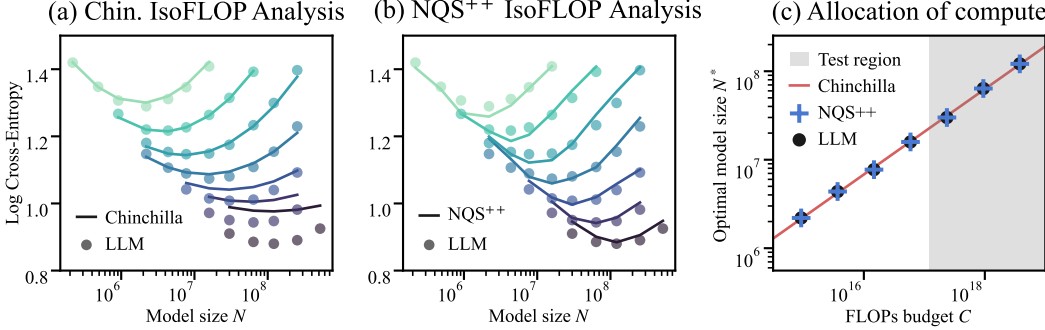

Figure 7: For LLMs trained with SGD, NQS$^{++}$successfully fitted the IsoFlop curves and matched Chinchilla and ground truth in resource allocation.

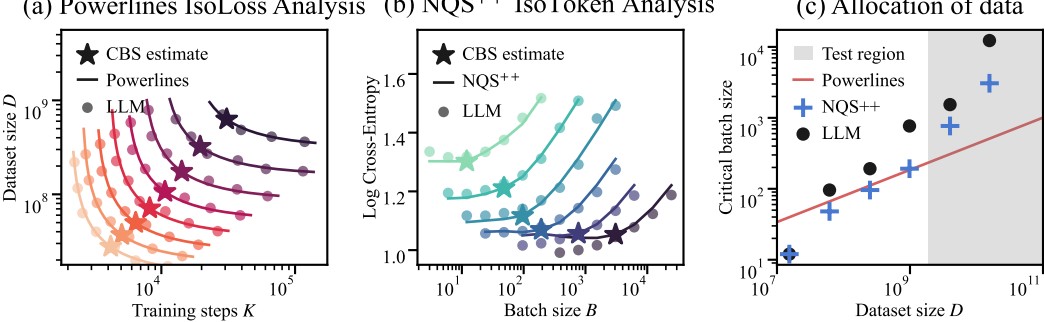

Figure 8: For LLMs trained with SGD, NQS$^{++}$successfully fitted the IsoToken curves and chose critical batch sizes (CBS) that are close to the ground truth. *Important Note*: CBS selected by Powerlines is not expected to match LLM, because the LLM points in (c) used the NQS$^{++}$version of CBS definition.

### E.4 Fitting NQS to LLMs Trained with Adam And a Cosine Learning Rate Schedule

Table 5: On LLMs trained with Adam and a cosine learning rate schedule, NQS$^{++}$outperformed Chinchilla on extrapolated compute budgets (IsoFlops), and explained 90% of the variance due to variation in batch sizes (IsoTokens). Note that on the IsoFLOPs test set, Chinchilla gave negative variance-explained values: this was due to the flatness of the IsoFLOP curves in the test set; the variance within each FLOPS budget was smaller than the squared difference between the LLM loss and the Scaling Model loss.

| Scaling Model | Add. train var. explained on | | Add. test var. explained on | |
| --- | --- | --- | --- | --- |
| | IsoFLOPs | IsoTokens | IsoFLOPs | IsoTokens |
| Chinchilla | **93** | - | −216 | - |
| NQS$^{++}$ | 83 | **95** | **74** | **92** |

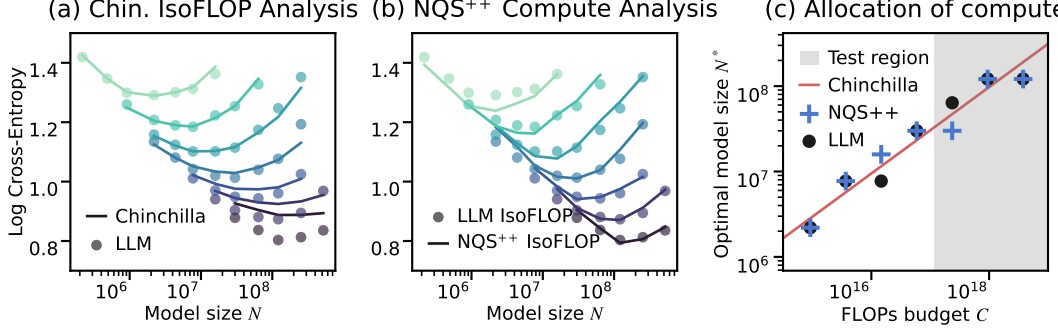

Figure 9: For LLMs trained with Adam and a cosine learning rate schedule, NQS$^{++}$successfully fitted the IsoFlop curves and matched Chinchilla and ground truth in resource allocation.

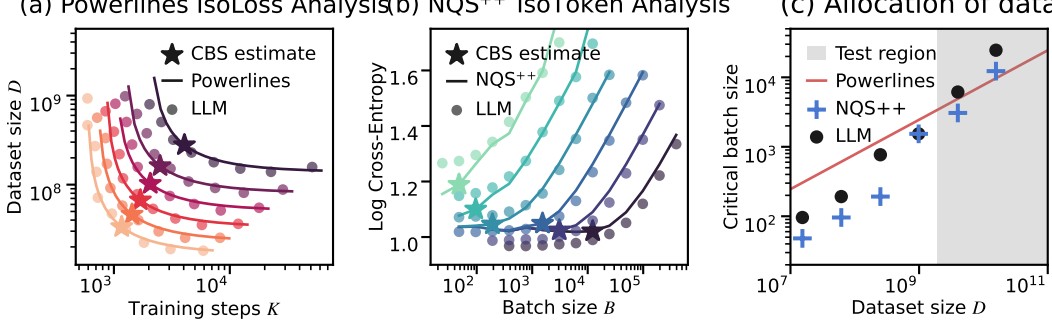

Figure 10: For LLMs trained with Adam and a cosine learning rate schedule, NQS$^{++}$successfully fitted the IsoToken curves and chose critical batch sizes (CBS) that are close to the ground truth. *Important Note*: CBS selected by Powerlines is not expected to match LLM, because the LLM points in (c) used the NQS$^{++}$version of CBS definition.

### E.5 FITTING CHINCHILLA ON THE HOFFMAN DATASET

Table 6: Chinchilla does not extrapolate well within the Hoffman dataset. As we removed the highest FLOP budget data points from its training data, Chinchilla's performance on training improved, but performance on the highest IsoFLOPs slice deteriorated. The dataset and Chinchilla fitting methodology is from Besiroglu et al. (2024) and we selected the IsoFLOPs subset for this analysis.

| | Add. var. explained | | Compute gap |
|---|---|---|---|
| Chinchilla fitted on | Train | Test (@3e21) | |
| IsoFLOPs $\leq$ 6e19 | **92** | −456 | 50x |
| IsoFLOPs $\leq$ 3e20 | 85 | −110 | 10x |
| All IsoFLOPs (max 3e21) | 80 | **3** | None |

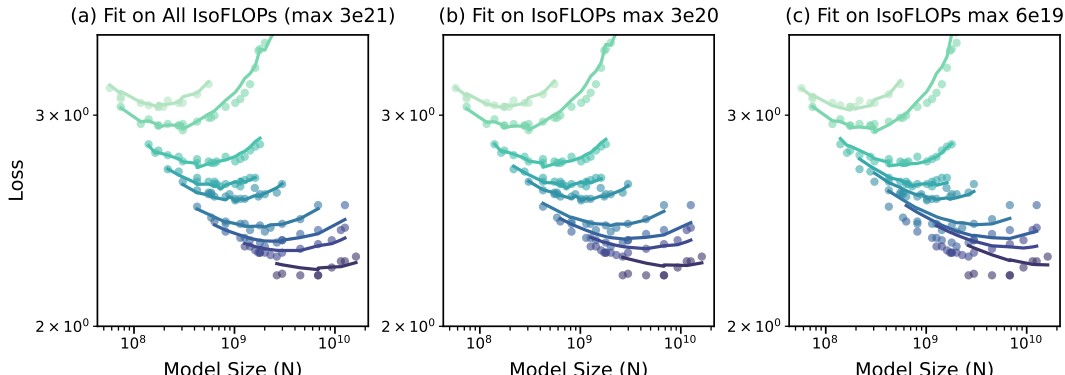

Figure 11: Chinchilla does not extrapolate well within the Hoffman dataset. As we removed the highest FLOP budget data points from its training data, Chinchilla's fit on the highest IsoFLOPs slice deteriorated. The dataset and fitting methodology is from Besiroglu et al. (2024) and we selected the IsoFLOPs subset for this analysis.

### E.6 THE MECHANISM OF LEARNING RATE ADAPTATION

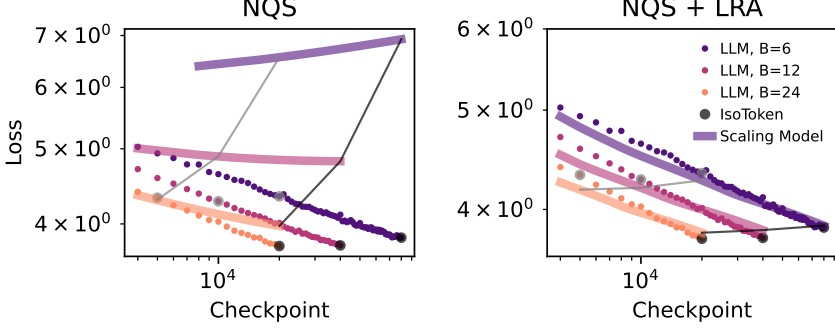

Figure 12: Learning Rate Adaptation (LRA) helped NQS++ match the training trajectory of LLMs. With LRA, the loss is approximately level between iso-token points (where batch size × number of batches processed is held constant); this is consistent with LLMs.

## E.7 SENSITIVITY TO THE NQS$^{++}$HYPER-PARAMETERS

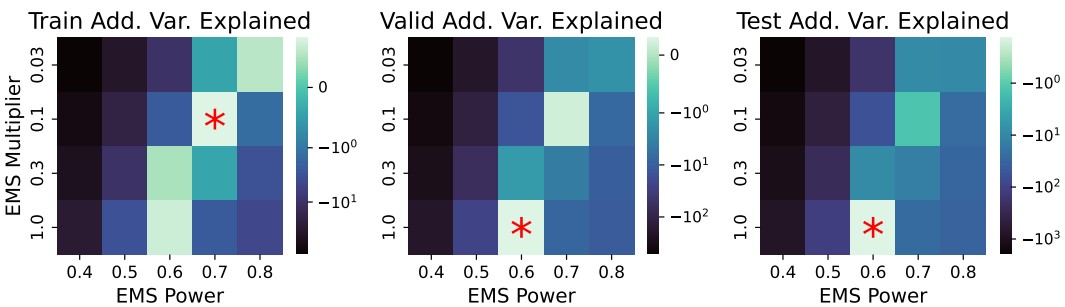

Figure 13: Sensitivity of the fit of NQS$^{++}$to the choice of the EPC parameters (power and multiplier), as measured by the Additional Variance Explained metric on the IsoFLOPs dataset. The red asterisk marks the cell with the highest Add. Var. Explained. The selection is based on the highest Additional Variance Explained on the the validation set. Fitted on SGD-trained LLMs.

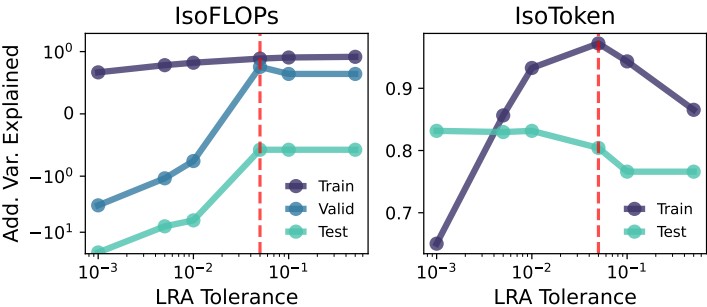

Figure 14: Sensitivity of the NQS$^{++}$fit to the choice of LRA tolerance, as measured by the Additional Variance Explained metric. The selected LRA tolerance, at 0.05, is marked by the red dashed line. The selection is based on the highest Additional Variance Explained on the the validation set (note that the validation set contains IsoFLOPs data points only). Fitted on SGD-trained LLMs.

## E.8 NQS SCALING PARAMETERS

Table 7: Comparison of the NQS$^{++}$scaling parameters for Adam and SGD, fitted on the training portion of our scaling datasets. $P, p$ are not directly comparable due to the different EMS hyperparameters. For $q$, the Adam value is smaller, likely reflecting better pre-conditioning properties. Adam-trained LLMs also appeared to have a smaller irreducible risk $\mathcal{E}_{\text{irr}}$, as inferred by the NQS. Interestingly, the fitted scaling exponent of the bias term $(p/q - 1/q)$ is comparable between the optimizers .

| Parameter | SGD | Adam |
|---|---|---|
| $p$ | 1.24 | 1.16 |
| $q$ | 1.21 | 0.89 |
| $P$ | 8.25 | 3.83 |
| $Q$ | 0.72 | 0.61 |
| $\sqrt{R}$ | 1.61 | 2.89 |
| $\mathcal{E}_{\text{irr}}$ | 1.07 | 0.31 |
| EMS $A$ | 1.00 | 0.10 |
| EMS $r$ | 0.58 | 0.70 |
| LRA Tolerance | 0.05 | 0.0001 |

# F  EXPERIMENT DETAILS

## F.1  LLM MODEL FAMILY

We define a model family as a function that maps a requested model size to a fully specified trainable model architecture. LLMs in the scaling datasets were trained with the GPT-NeoX suite in the Huggingface Transformers library (Wolf et al., 2020). In our experiments, the requested model sizes are of the form $1e6 \times 2^j$ for integers $j$, ranging from $0.25$ to $512$ million parameters. Due to the constraints of the model family, the actual achievable model sizes are not identical to the requested model size. Some of the constraints are: (1) for transformer models, the number of layers and hidden size are required to be integers, and the latter often multiples of 16; (2) we request a certain power law relationship between the number of layers, hidden size and the model size. In short, given a requested model size, we search for an LLM that is close to the requested size, and satisfies the constraints. Details are given below.

To construct the model family, we first fit a power law relationship on the existing Pythia suite of models (Biderman et al., 2023), by running regressing the hidden size ($H$) and the number of layers ($L$) against the model size ($N$):

$$\log(H) \sim p_H \log(N) + a_H, \text{ and } \log(L) \sim p_L \log(N) + a_L.$$

In the pythia family, the intermediate size is always four times the hidden size, and we follow that convention in our model family. We also define the number of heads to be hidden size$/16$. In Pythia the divisor is $\geq 64$. We chose 16 for convenience, so that we can have an integer number of heads as long as the hidden size is divisible by 16, and be able to construct smaller LLMs that closely match requested model sizes.

Given a requested model size $N_{\text{request}}$, we search in a neighborhood of $N_{\text{request}}$ (10% to 150%), for a value $N'$ that minimizes the difference:

$$\left| N_{\text{NeoGPT}}\Big( H = 16 \times \text{int}(\exp(p_H \log N' + a_H)/16), L = \text{int} \exp(p_L \log N' + a_L) \Big) - N_{\text{requested}} \right|.$$

Here $N_{\text{NeoGPT}}(H, L)$ denotes the count of trainable parameters of a GPT-NeoX LLM constructed with the given hidden size $H$ and number of layers $L$. Said constructed model is the output of the model family mapping for input $N = N_{\text{NeoGPT}}(H, L)$. Where possible, we prefer to use $N = N_{\text{NeoGPT}}(H, L)$ over $N_{\text{request}}$.

## F.2  SCALING DATASETS

**IsoFLOPs Dataset**. The IsoFLOPs dataset consists of 7 levels, each level contains LLMs trained with a fixed FLOP budget $C$, but with various $N/D$ allocation (by default, we use the Powerlines critical batch size to allocate $D$ to $B, K$). The FLOP budget quadruples between levels, resulting in an overall compute gap of $\times 4^6$. The first 4 levels are used for training (included in the computation of $\mathcal{L}_S$), level 5 is used as a validation set to select the EMS hyperparameters of NQS$^{++}$ as well as the tolerance of LRA, and the last 2 levels with the highest $C$ are reserved for testing. The validation and test data points in the IsoFLOPs dataset are from a small range around the optimal $N, D$ allocation. All included, the range of compute budget for the IsoFLOPs dataset is $9e14$ to $4e18$ FLOPs.

**IsoTokens Dataset**. The IsoTokens dataset is obtained by training LLMs at a fixed model size , and consists of 6 levels of data points, each level containing LLMs trained at a fixed number of tokens (fixed $D$, varying $B, K$). Between levels, $D$ quadruples, resulting in a $\times 4^5$ gap between the lowest and the highest levels. The first 4 levels are used for training, and the last 2 levels with the highest token counts are reserved for testing. All included, the range of compute budget for the IsoFLOPs dataset is $9e14$ to $9e17$ FLOPs.

## F.3  LLMS TRAINED WITH SGD

The experiment set up for the SGD trials were identical to that of the Adam trials, with the following exceptions: the LLMs were trained with an SGD optimizer with a learning rate of $1.999$. We chose this learning rate because in our experiments this was nearly optimal on the range of LLMs we tested.

