# OpenReview forum: "Noisy Quadratic Models of Scaling Dynamics"
_ICLR.cc/2026/Conference — Submitted to ICLR 2026_

### Official Review · Reviewer_GEC9 · 2025-10-28

**Soundness:** 2
**Presentation:** 1
**Contribution:** 3
**Rating:** 4
**Confidence:** 2

**Summary:**

This paper investigates more sophisticated theory-inspired scaling law models for LLM training that can predict the loss over the batch size, model size and dataset size. The addition of the batch size is an improvement over prior scaling laws like those from Chinchilla. Their model, the Noisy Quadratic System (NQS) is based on modelling LLM training as stochastic gradient descent (SGD) on an infinite-dimensional quadratic loss function, projected onto its top N eigendirections. On top of this theoretically inspired model, the authors propose two more empirical adjustments, creating NQS++. The Effective Model Size adjust the parameter count to improve the overall fit, and Learning Rate Adaptation which simulates changing the learning rate which is needed to fit small batch sizes better. Experiments are performed on Pythia-style models and claim improvements over prior approaches like Chinchilla, for example when predicting test losses on out-of-sample compute budgets.

**Strengths:**

* Aims to bridge theory and practice, from purely empirical scaling laws to theoretical models that are hard to fit.
* Incorporation of the batch size which prior scaling law approaches had a hard time with.
* Strong empirical performance on their experiments.
* The authors show how to make their scaling models computationally efficient which they are otherwise not when using naive approaches.
* Upfront about limitations.

**Weaknesses:**

* The paper feels poorly presented. The authors assume the reader is deeply familiar with related literature and frequently cite it without summarizing. There are also numerous references to the appendix without summarization. Overall the manuscript does not feel very self-contained and is likely relatively inaccessible to a large fraction of the community. I think the work could be significantly improved by better explaining the core concepts at a level that is accessible to a broader audience. Without this the impact of the work is significantly reduced despite presenting interesting and promising ideas.
* Reliance on additional empirical fixes. The core theoretical model does not seem to be powerful enough to model the dynamics of LLM training accurately. The authors introduce additional tricks to deal with this but that deviates from the principled theoretical approach that inspired the paper.
* The experimental approach has some key limitations. Making general claims about batch size optimality without tuning the learning rate for each batch size is not principled. The same goes for the beta2 and to a lesser degree the beta1 coefficients of Adam. The approach does not seem to generalize across learning rates or optimizers as the authors acknowledge in their limitations.

**Questions:**

The claims about batch size optimality seem limited by the fixed LR experimental setup. How would you expect the NQS parameters to behave in a more realistic setting where the learning rate is properly co-tuned with the batch size? Would the LRA fix still be necessary in that case?

---

> ### Author Response · Authors · 2025-11-22
>
> We thank the reviewer for their insightful observations and detailed feedback.
>
> ---
>
> ## **Response to Weaknesses**
>
> > ### The paper feels poorly presented. The authors assume the reader is deeply familiar with related literature and frequently cite it without summarizing.
>
> In the second revision to the draft, we re-organised the introduction section, and added Sec. 2.2 to describe the related theoretical scaling models in details.
>
> Thanks again for taking the extra time and efforts to understand our draft!
>
>
> > ### Reliance on additional empirical fixes. The core theoretical model does not seem to be powerful enough to model the dynamics of LLM training accurately. The authors introduce additional tricks to deal with this but that deviates from the principled theoretical approach that inspired the paper.
>
> EMS is indeed heuristic-based; a future direction is to look for evidence to support this modification. LRA, however, is corroborated by theory and highly mechanistic.
>
> Although our implementation of LRA is novel, we are not the first to propose an adaptive algorithm as a theoretical explanation for LLM training behaviours. McCandlish et al. (An Empirical Model of Large Batch Training) suggests that a quadratic optimization, with line search over the learning rate, models the LLM loss profiles with changing batch size. The LRA algorithm can be seen as a crude approximation of their hypothesis. For the mathematical motivations, please see Appendix D.1 of McCandlish et al.
>
> We provided a new figure (Appendix E.6) displaying the loss trajectory of NQS with and without LRA, compared with LLM checkpoints. The figure illustrates that incorporating the LRA allows the NQS to accurately predict the loss trajectory of LLMs during training. This cannot be achieved by simple curve-fitting on the IsoToken curves.
>
> We believe that LRA is highly mechanistic, and serves to highlight the merits of our rolled-out optimization approach: we posed a straightforward modification on the “effective optimizer” of LLM training (namely a greedy line search). This level of intervention is not possible without a mechanistic model.
>
>
>
>
> > ### Making general claims about batch size optimality without tuning the learning rate for each batch size is not principled. The same goes for the beta2 and to a lesser degree the beta1 coefficients of Adam. The approach does not seem to generalize across learning rates or optimizers as the authors acknowledge in their limitations.
>
> We thank the reviewer for noting that co-tuning of learning rate and batch size is important in many practical settings. So far, the NQS does not seem to work well with mixed learning rate, and we do not know how it would perform in the co-tuned set up. However, given its kinship to NQM (Zhang et al., 2019), which provides qualitative results consistent with LLMs in the co-tuned setting, it seems to be a promising direction to explore.
>
> Thank you for directing us to the idea of co-tuning beta1, beta2 of the Adam optimizer. In the revised draft, we added a discussion of recent literature that incorporates the tuning of beta1 and beta2 (Marek et al. 2025) in the extended related works section.  Co-optimality of optimizer and hyper-parameters is a subtle question: the optimizer can have a great impact on the optimal batch size. We suspect the influence on critical batch size (i.e. time-constrained setting) is smaller but we have not yet conducted experiments in this direction.
>
> > ### The approach does not seem to generalize across learning rates or optimizers as the authors acknowledge in their limitations.
>
> In practice, this lack of generalisation is a lesser concern, as long as the modeller could obtain training runs of LLMs that use similar optimizer and learning rate configurations as the target test run.
>
> As a clarification to the limitation that we acknowledged: we found that NQS++ trained on LLM runs with the SGD optimizer cannot provide good predictions for LLMs trained with Adam. However, as we demonstrated with experiments on SGD (App. E.3) and Adam (Tab. 2, Fig.1 & 2), the NQS++ can be applied to multiple optimizers.
>
> **(To be Continued in the Next Comment Box)**

---

> > ### Author Response · Authors · 2025-11-27
> >
> > (Continued)
> >
> > ## **Answer to Question**
> >
> > >  ### The claims about batch size optimality seem limited by the fixed LR experimental setup. How would you expect the NQS parameters to behave in a more realistic setting where the learning rate is properly co-tuned with the batch size? Would the LRA fix still be necessary in that case?
> >
> > We do not know if the NQS would perform well where LR is co-tuned with batch size, as NQS does not predict well on datasets with mixed initial learning rate values. We do think that this is the most important limitation of our work.
> >
> > We agree that the effect of LRA would be dampened at the optimal learning rate for a given batch size. However, because the loss at sub-optimal learning rates cannot be predicted without LRA, LRA is necessary if we want NQS to aid in the selection of the optimal LR. Therefore, we expect that LRA is still useful in the co-tuned setting.
> >
> > At a higher level, LRA helps NQS more closely mimic the training trajectory of LLM runs, and is likely to be useful in broader settings (for a visualisation, please check out the new figure in Appendix E.6).

---

### Official Review · Reviewer_maGC · 2025-11-01

**Soundness:** 3
**Presentation:** 3
**Contribution:** 3
**Rating:** 6
**Confidence:** 3

**Summary:**

This paper introduces the Noisy Quadratic System (NQS), a parametric model for simulating the scaling dynamics of LLM losses during pre-training. By extending theoretical noisy quadratic models, NQS incorporates model size, batch size, and training steps, enabling predictions of optimal training configurations under various resource constraints like compute, memory, and time.

To improve predictive accuracy, they also introduce NQS++, incorporating Effective Model Size (EMS) to correct for loss curvature discrepancies and Learning Rate Adaptation (LRA) to adjust for overestimation at small batch sizes. Experiments on small-scale Pythia models show that NQS++ can extrapolate to compute budgets 64× larger than training data, outperform Chinchilla, explaining 90% of batch size variance.

**Strengths:**

The proposal tries to bridge theoretical scaling models (Zhang et al., 2019; Bahri et al., 2021) with practical applicability, allowing a single model to handle multiple decisions like batch size optimization, which Chinchilla ignores. Novelty arises from making NQS fittable via simplifications (e.g., deterministic projections) and numerical approximations (e.g., Euler-Maclaurin for sums), plus extensions like Effective Model Size (EMS) and Learning Rate Adaptation (LRA) to fix observed mismatches in curvature and small-batch behavior.

Experiments are rigorous, using IsoFLOPs and IsoTokens datasets for fitting, with clear metrics (e.g., additional variance explained ≥85-90% out-of-sample). The inclusion of a variance term for batch size effects is another key contribution for practitioners under time/memory constraints, and the model's extensibility (e.g., to batch schedules) adds impact.

**Weaknesses:**

The core quadratic assumption, while simplifying inference, may not capture real LLM dynamics fully, as the paper also acknowledges. Experiments are limited to small models (<500M parameters) and a specific architecture, the contribution could be strengthened if discussions/experiments are conducted to address generalization to frontier-scale LLMs, multi-epoch training, or diverse optimizers/datasets.

The single-epoch noise independence might overlook data reuse effects seen in recent works (Lin et al., 2025a). Fitting relies on non-convex optimization with parallel initializations, but lacks sensitivity analysis to hyperparameter choices (LRA tolerance). While outperforming Chinchilla, baselines are limited; comparisons to more recent batch-aware scaling laws (Bi et al., 2024; Bergsma et al., 2025) could strengthen claims.

**Questions:**

How does NQS generalize to multi-epoch training or data reuse, given the independence assumption in noise? Could you discuss extensions or limitations relative to works like Lin et al. (2025b)?

For LRA, the greedy approximation adds compute—how does its cost scale with K, and could you compare to full adaptation empirically?

The paper mentions failure modes; in what regimes (high compute) does NQS deviate most from LLMs, and how might this inform future extensions?

---

> ### Author Response · Authors · 2025-11-22
>
> We appreciate the reviewer's positive assessment and thoughtful feedback.
>
> ---
> ### **Response to Weaknesses**
>
>
> > ### Fitting relies on non-convex optimization with parallel initializations, but lacks sensitivity analysis to hyperparameter choices (LRA tolerance).
>
> We added the requested sensitivity analysis to the EMS (power, multiplier) and LRA (tolerance) in App. E.7. The figures show that hyper-parameter choices based on the validation set transfer well to the test set.
>
> > ### Experiments are limited to small models (<500M parameters)
>
> We are actively looking for resources to evaluate our model on larger LLM and more diverse runs. We would greatly appreciate it if the reviewers could provide suggestions on what would be a reasonable LLM flop budget for scaling analysis.
>
> > ### While outperforming Chinchilla, baselines are limited; comparisons to more recent batch-aware scaling laws (Bi et al., 2024; Bergsma et al., 2025) could strengthen claims.
>
>
> We agree a comparison to other batch-size selection methodologies would strengthen the claims; however, the use case of these methods do not match our sufficiently so as to make comparison possible. More specifically:
>
> - In Bergsma et al.'s method, the constraint is a certain target loss, and the method looks for the least time/compute intensive training configurations that achieves this loss. In our methods, the constraint is a fixed time/compute budget, and the method looks for the training configurations that gives the lowest loss possible. Therefore, Bergsma et al. addressed the dual of our problem, but it is not directly comparable as a benchmark without extensive modificaiton.
>
> - Deepseek looks for the optimal batch size (i.e. not time constrained); while our methods looks for the critical batch size (i.e. time constrained).
>
>
>
>
>
> ### **Answer to Questions**
>
> > ### How does NQS generalize to multi-epoch training or data reuse, given the independence assumption in noise?
>
> Thank you for the insightful question. Our paper only considered the single-epoch case. In order to extend this work to multi-epoch training, we do expect some assumptions on the correlations between noise would be necessary.
>
> > ### Could you discuss extensions or limitations relative to works like Lin et al. (2025b)?
>
> In the second revision to our draft, we included a background section (Sec. 2.2) that discussed theoretical works like Lin et al. and its relation to our work. We summarise the discussion below:
>
> - Lin et al. (2025b) assumed an exponentially decaying learning rate schedule, which results in a negligible variance term in the asymptotics. Therefore Lin et al. is not suitable for analysing batch size.
> - Paquette et al., to our understanding, discussed case-by-case the implication of noise under constant learning rate schedule and varying degrees of noise. However, the fact that we need the LRA elaboration to the NQS shows that LLM training (even at fixed lr) is not well-captured by a quadratic function optimized at fixed learning rate.
> - The line of work by Bordelon et. al similarly did not address noise explicitly, with asymptotic results mostly concerning the training horizon and model size.
>
> > ### For LRA, the greedy approximation adds compute—how does its cost scale with K, and could you compare to full adaptation empirically?
>
> The cost of LRA is log(K) (for a proof sketch, please see Appendix B.3).  In our experiments, evaluating NQS+LRA takes about 1 second, with each vanilla NQS evaluation taking about 0.001 seconds.
>
> Compared to vanilla NQS, NQS+ LRA is more expensive due to a constant factor that is proportional to the number of change points allowed (i.e., at most 100 LR updates are allowed in our analysis) and maximum number of search trials at each change point (about 5 is sufficient for our analysis).
>
> We expect the cost of full adaptation to be exponential in the number of change points. We expect this to take at least minutes to compute so we did not test this empirically.
>
> > ### The paper mentions failure modes; in what regimes (high compute) does NQS deviate most from LLMs, and how might this inform future extensions?
>
> One regime where NQS deviates from LLMs is the effect of learning rate. The NQS does not seem to generalize across the learning rate dimension: one cannot train NQS++ on one learning rate and expect good predictions on LLMs trained at a much larger learning rate.
>
> In theory, the scaling parameter Q can absorb changes in learning rate (γ). A priori, we suspected that one could increase Q to predict the LLM’s response to an increase in γ. However, we empirically observed that LLMs were less sensitive to changes in γ than our quadratic system, and adjusting Q by the same ratio tend to over-state the impact. Thus, an important direction for future work is to investigate how the NQS can be extended to accurately model the effects of changing learning rate.

---

### Official Review · Reviewer_GytH · 2025-11-02

**Soundness:** 3
**Presentation:** 2
**Contribution:** 3
**Rating:** 4
**Confidence:** 3

**Summary:**

The authors propose the Noisy Quadratic System (NQS), a fittable, 6-parameter model of scaling dynamics inspired by theoretical quadratic models, to model scaling laws for LLMs. The NQS models the final loss $L(N, B, K)$ by decomposing it into three terms: an approximation error $E_{app}(N)$, a bias term $E_{bias}(N,K)$, and a variance term $E_{var}(N,B,K)$, where $N,B,K$ are model size, batch size, and training steps. To make NQS work well, the authors extend it to NQS++ by adding two empirical components, "Effective Model Size" (EMS) and "Learning Rate Adaptation" (LRA), which correct for mismatches between the simple quadratic model and real LLM losses.

The central claim is that NQS++, when fit on small-scale experiments, can accurately extrapolate to larger compute budgets where the Chinchilla model fails. Because NQS++ models $B$ and $K$, it can be used to optimize for complex, compound constraints (e.g., compute+memory) and even rank the performance of different batch size schedules.

**Strengths:**

* Originality: The paper's main contribution is essentially bringing together two directions: the practical, empirical scaling-law fitting (like Chinchilla) and the theoretical, mechanistic models of scaling. It bridges those two with noisy quadratic models, which were successful in the NN optimization literature before. By making a simplified theoretical model fittable, the authors create a "semi-mechanistic" surrogate that is more powerful and flexible than a simple power-law formula.

* Quality and significance: The paper is generally done solid, with lots of details and derivations in the appendix. If the model's utility holds, it would allow better extrapolation, application to real-world constraints (like memory), or better understanding of scaling dynamics. The fitting results seem compelling. Moreover, the ability to evaluate batch size schedules (Fig 4) is a novel capability that, to my knowledge, is new.

* Clarity: The main ideas are relatively clearly conveyed by the paper (though I have some comments below). The experimental design, built around IsoFLOPs and IsoTokens setups, provides a clear framework for evaluating the model's interpolation and extrapolation capabilities, with well done figures.

**Weaknesses:**

* The "++" extension feels ad-hoc: The base NQS model, which is principled, basically fails (Table 2). The model's success is entirely dependent on the NQS++ extensions, which are essentially empirical patches. The EMS $N_{eff}=(AN)^r$ is just a new power law, and LRA is a greedy algorithm based on a hypothesis about normalization layers. This reliance on curve-fitting "patches" somewhat contradicts the "mechanistic" claim of the model.

* Related to the above, applying the model (judging purely from the text) seems rather complex with the extensions. What made the empirical scaling laws (like Chinchilla) so attractive is also their simplicity. Perhaps this can be remedied if the authors release an easy to use package for future use.

* Experimental scope: The authors admit this, but it's one of the obvious weaknesses. All claims are based on one model family on one dataset at a small scale ($<10^{19}$ FLOPs). The claim that Chinchilla "overfits" is hard to justify; how much effort was put into working on and improving the Chinchilla fits? Similar efforts to making the NQS and its extension work? It's also possible that the Chinchilla functional form is not a good fit at this scale or would fit differently with other setups, and the arguably more complex NQS++ model is just a better fit here.

* Writing: It's clear upon reading the paper that the authors are knowledgeable and have worked a lot on this project. However, this may have also lead to too dense information and a lack of flow. For example, the introduction contains lots of paragraphs concatenated together, before any of the core concepts are introduced formally, with a sort of early conclusion. Then throughout, some key points emerge abruptly; providing more explicit framing would enhance readability. For example, the NQS extension or computational complexity seemingly come out of nowhere (e.g., since experiments only follow 2 pages after). In short, the writing would strongly benefit from transitional sentences, paragraphs, or overview descriptions.

**Questions:**

Perhaps all questions are in the section above, but to potentially more direct: How much effort was put into making Chinchilla work better, i.e., how 'tuned' is the baseline? Similarly, how much trial and error was involved in the EMS and LRA extensions?

Moreover, NQS++ must be tuned on a validation set. This validation set uses runs "at least 4 times larger" than the training runs. How sensitive is the final model's accuracy to the choice of this validation set? Does this requirement partially negate the "train on small-scale" promise?

---

> ### Author Response · Authors · 2025-11-22
>
> We appreciate the reviewer’s careful reading of our paper and their insightful suggestions.
>
> ---
>
> ### **Answer to Questions and Response to Weakness**
>
> **On Chinchilla**
>
> >  ### It's possible that the Chinchilla functional form is not a good fit at this scale or would fit differently with other setups, and the arguably more complex NQS++ model is just a better fit here.
>
> We agree the ideal set up to understand if our model outperforms Chinchilla is one that is comparable to the Hoffman set up.
>
> In a set of new experiments (Appendix E. 5), we show that Chinchilla failed to extrapolate towards a higher compute budget even on the original Hoffman dataset (IsoFlops portion) , consistent with our small LLM runs. When all data points are used to obtain the Chinchilla parameters, Chinchilla was able to fit the highest-flop slice very well, as presented in the Hoffman paper; however, as the highest-flop slice is removed from training, the fit on the highest compute slice deteriorates. Unfortunately, we could not run NQS++ on the original Hoffman data due to the lack of Batch size information, but we hope the above improves confidence in our small-scale LLM dataset.
>
> As originally presented in Hoffman, the use case of Chinchilla is to predict optimal N/D allocation. In our experiments, we show that Chinchilla is good at this task, but we evaluated Chinchilla on tasks that were not in scope of the Hoffman paper.
>
>
> > ### The claim that Chinchilla "overfits" is hard to justify; how much effort was put into working on and improving the Chinchilla fits?
>
> Here are some further details on how we fitted Chinchilla:
>
> 1. We used the method from Besiroglu et al. (2024), which is improved over the original Chinchilla methodology.
> 2. The Chinchilla fitting procedure has an initialization range that is tunable, but given the near-perfect performance on the training set, i.e. a reasonable solution was found within the range, we did not modify the range.
> 3. The optimization procedures for Chinchilla could potentially be improved (i.e. the L-BFGS may not be the best optimizer and Huber loss may not be the best loss function), but we did not look into this.
>
> We believe the near-perfect fit on the training set but worse performance on the test set shows an inherent weakness of the Chinchilla functional form, which cannot be addressed by optimising the fitting procedure.
>
> **On the ++ Extensions**
>
> >  ###  The EMS is just a new power law, and LRA is a greedy algorithm based on a hypothesis about normalization layers. This reliance on curve-fitting "patches" somewhat contradicts the "mechanistic" claim of the model.
>
> We are also somewhat bothered by the form of the EMS elaboration, which is heuristic-based; a future direction is to look for evidence to support this modification. LRA, however, is corroborated by theory and highly mechanistic.
>
> Although the implementation is new, we are not the first to propose an adaptive algorithm as a theoretical explanation for LLM training behaviours. McCandlish et al. (An Empirical Model of Large Batch Training) suggests that a quadratic optimization, with line search over the learning rate, models the LLM loss profiles with changing batch size. The LRA algorithm can be seen as a crude approximation of their hypothesis. For the mathematical motivations, please see Appendix D.1 of McCandlish et al.
>
> We provided a new figure (Appendix E.6) displaying the loss trajectory of NQS with and without LRA, compared with LLM checkpoints. The figure illustrates that incorporating the LRA allows the NQS to accurately predict the loss trajectory of LLMs during training. This cannot be achieved by simple curve-fitting on the IsoToken curves.
>
> We believe that LRA is highly mechanistic, and serves to highlight the merits of our rolled-out optimization approach: we posed a straightforward modification on the “effective optimizer” of LLM training (namely a greedy line search). This level of intervention is not possible without a mechanistic model.
>
>
>
> **(To be continued in the next comment box)**

---

> ### Author Response · Authors · 2025-11-27
>
> ### **Answer to Questions and Response to Weakness**  **(Continued)**
>
> ------
> **On the ++ Extensions**
>
> > ### how much trial and error was involved in the EMS and LRA extensions?
>
> Indeed some efforts were invested to find EMS and LRA.
>
> For EMS, we notice that without this adjustment the curvature of the IsoFLOP curves are off, and the optimization of the scaling loss frequently diverges; so we started with a flat factor that discounts the parameters, then a pure power law, but in the end we decided to use a power law with both slope and intercept.
>
> For LRA, this took less trial and error. It was clear that the variance term of the quadratic model needs to be further controlled and an adaptive learning rate seemed natural; the results is not very sensitive to the particular implementation of the algorithm.
>
> For a fair comparison with Chinchilla, we only looked at the test slices once the NQS++ method were finalized.
>
> **On the validation set**
> >  ### The validation set uses runs "at least 4 times larger" than the training runs. How sensitive is the final model's accuracy to the choice of this validation set? Does this requirement partially negate the "train on small-scale" promise?
>
> Because the test data is 4 to 16 times larger in FLOPs than the validation set (x64 larger than training), our experiments supports the claim that NQS++ trained on small scale can make prediction on larger scale LLM runs.
>
> We added a new experiment (App. E7) showing the sensitivity of the NQS fit to the hyper-parameters tuned on validation set. Finding the optimal hyper-parameters required fitting on the validation set: the selection on the validation set tends to coincide with the best choice on test, while the optimal hyper-parameters based on the training set do not work. The scale of the validation set is chosen to be between train and test, so as to capture trends along the FLOPs scales.
>
> **Ease of Use**
> >  ### Applying the model (judging purely from the text) seems rather complex with the extensions. What made the empirical scaling laws (like Chinchilla) so attractive is also their simplicity. Perhaps this can be remedied if the authors release an easy to use package for future use.
>
> Thank you for the suggestion! We do plan to release an easy to use package. Fitting NQS should be similar to fitting any other statistical model that requires tuning on a validation set. The search for the EMS and LRA parameters is reasonably fast, with each tuning run taking about a minute; we plan to include utilities for the user to define their own validation metric and run sensitivity testing.
>
> **Writing**
> >  ### The introduction contains lots of paragraphs concatenated together, before any of the core concepts are introduced formally, with a sort of early conclusion. Then throughout, some key points emerge abruptly; providing more explicit framing would enhance readability.
>
> In the second revision to the draft, we re-organized the introduction and preliminary sections: the Chinchilla and theoretical models are first introduced, and comments on them are delayed until after the introduction; we also make connections between the models where appropriate.

---

### Official Review · Reviewer_8D3C · 2025-11-04

**Soundness:** 2
**Presentation:** 2
**Contribution:** 2
**Rating:** 4
**Confidence:** 3

**Summary:**

This paper proposes Noisy Quadratic System (NQS and NQS++), which are new functional forms for LLM scaling laws as a function of model size, batch size, and training steps, and can use dataset size, training flops, and peak memory as constraints. Their models act as an alternative to the Chinchilla scaling model (Approach 3) or other scaling laws based on high-dimensional linear regression / random features models. In particular, NQS includes a variance term intended to capture batch size effects. NQS++ extends the basic NQS by adding two empirically-motivated extensions to improve curvature modeling and small batch behavior.

They assume the LLM optimizes a quadratic loss function with power-law distributed eigenvalues and noise, giving an expression with three terms (approximation error, bias and variance) and six fitted parameters.

They evaluate on Pythia-style models up to 500M parameters trained on OpenWebText2. NQS++ outperformed Chinchilla on explaining the variance on test sets with a large (64x) compute gap from training data, and explained variance due to batch size changes (which Chinchilla can’t do at all). NQS++ was also predictive of close-to-optimal configurations under various resource constraints (time, memory, data) not just compute-optimal. Chinchilla overfit on their scaling dataset whereas NQS++ extrapolated well.

**Strengths:**

This is a worthwhile problem to tackle: the functional form of the Chinchilla scaling laws is almost certainly wrong (the asymptotics don’t make sense), and fails to capture batch size effects.

**Weaknesses:**

I found the motivation in the paper for the particular functional form in NQS/NQS++ hard to follow. Models with additional terms and more fitted parameters are likely able to fit scaling data better than simpler models like Chinchilla, but the evidence is somewhat weak that this functional form is worth the additional complexity and likely to generalize well. The experiments were done with constant learning rates and on relatively small compute scales (<10^19 flops) which makes it hard to tell how practical the results are.

**Questions:**

N/A

---

> ### Author Response · Authors · 2025-11-22
>
> We thank the reviewer for their thoughtful comments and constructive feedback.
>
>
> -----
> >  ### I found the motivation in the paper for the particular functional form in NQS/NQS++ hard to follow.
>
> The functional form is inspired by the theoretical scaling models.
>
> In the revised draft, we added a new section (2.2 background on theoretical model) to provide motivation to the NQS.
>
> -----
>
> >  ### The experiments were done with constant learning rates and on relatively small compute scales (<10^19 flops) which makes it hard to tell how practical the results are.
>
> Based on the reviewer's concern over the constant learning rate schedule, we have added experiments of NQS++ fitted on LLMs trained with a cosine learning rate schedule (appendix E.4). The performance is comparable with that of NQS++ on a constant learning rate schedule.
>
> We are currently exploring resources that would allow us to evaluate our model on larger-scale LLMs. We would be grateful if the reviewers could advise on what they consider an appropriate FLOP budget for scaling analysis.
>
> Without the additional resources, we believe the merit of a scaling model can be assessed by how well the model extrapolates across scales, which NQS++ succeeded in. The LLMs in our test set are up to x64 times larger than the largest LLM in the training set, and x16 times larger than the largest run in the validation set; the full data set expands x1000 folds in flops, a range that is comparable to the original Chinchilla (Hoffman et al.) dataset. This gives us some confidence that NQS++, if given training data at a greater scale, would be capable of making decisions on frontier models at the Chinchilla level.
>
> ---------
>
>
> >  ### Models with additional terms and more fitted parameters are likely able to fit scaling data better than simpler models like Chinchilla, but the evidence is somewhat weak that this functional form is worth the additional complexity and likely to generalize well.
>
>
> We agree with the reviewer that NQS++ is more elaborate than Chinchilla, and it is very important for us to understand if NQS overfits. We conducted training/validation/holdout testing, the standard statistical approach to handle the comparison of models with potentially different levels of complexity. Within the limited scope of our experiments, there is no evidence that our model class failed to generalize to the test slices.
>
> The EMS elaboration is indeed heuristic-based; but we can provide further motivation for the LRA: although the implementation of LRA is new, we are not the first to propose an adaptive algorithm as a theoretical explanation for LLM training behaviours. McCandlish et al. (An Empirical Model of Large Batch Training) suggests that a quadratic optimization model, with line search over the learning rate, closely models the LLM loss profiles with changing batch size. Our LRA algorithm can be viewed as a crude approximation of their hypothesis. For the mathematical motivation, see Appendix D.1 of McCandlish et al., 2018.
>
> We made a new figure (Appendix E.6) with the training trajectory of LLMs overlaid with vanilla NQS and NQS + LRA. NQS + LRA tracks the LLMs closely during the training process, which is not achievable with the vanilla version.
>
> Despite the apparent complexity, because the NQS++ model class is highly structured, it is not a particularly large model class, and in particular not more complex once Chinchilla is combined with additional heuristics approaches (e.g. Bergsma et al.) to model the batch size and time dimensions.
>
> We recognize that despite the rigorous out-of-sample testing, we have developed NQS++ on a very limited (dataset + architecture) combination. Given more resources, we would love to run more experiments to test our understanding.

---

### Author Response · Authors · 2025-11-23
**Summary of the Response**

We sincerely thank all four reviewers for their thoughtful feedback.

### **Summary of Reviews and Our Response**

The reviewers are positive about the potential impact of this work if the results were to generalize: “If the model's utility holds, it would allow better extrapolation, application to real-world constraints (like memory), or better understanding of scaling dynamics” (reviewer GytH), and  “the model's extensibility adds impact” (reviewer maGC31).  The reviewers found our methodology novel, and our experiment design rigorous, with strong empirical performance on the experiments conducted.

However, there are a few areas of reservations, mostly leading to uncertainties about the generalisation of our model.

**Concern #1**:

The reviewers are concerned that the success of our scaling model may not transfer to larger LLMs. E.g., whether it is an artifact of our smaller LLMs setup (<1e19 FLOPs) that NQS++ outperformed Chinchilla.

**Our Response**:

We would be grateful for guidance on an appropriate FLOP budget, as we are seeking resources to run larger-scale experiments.

Alternatively, we believe the merit of a scaling model can be assessed by how well the model extrapolates across compute scales. In our experiments, NQS++ extrapolated successfully up to x64 in compute (outperforming Chinchilla). Our dataset expands x1000 folds in FLOPs, comparable to the Chinchilla (Hoffman et al.) dataset. This makes us confident in NQS++'s ability to make decisions on frontier models at the Chinchilla level, once given the appropriate data.

Our new experiments (App. E.5) showed that Chinchilla failed to extrapolate over the Hoffman et al. dataset, consistent with results on our small LLM runs.

**Concern #2**:

Our experiments did not use the latest learning rate schedule and optimizers (but only SGD and Adam at constant learning rate); batch size is not co-tuned with learning rates.

**Our Response**:

We agree broader experimentation would strengthen the paper: in App. E.4, we present new experiments on cosine learning rate schedules.

In its current state, NQS++ trained with one protocol (e.g. SGD at constant LR) is not capable of making predictions on LLMs trained with another protocol (e.g. Adam with co-tuned LR), but this should not preclude practical use of the NQS, as long as the training LLMs can be designed to match the target LLM. Indeed, the Hoffman paper relied on a fixed set of heuristics for selecting batch size and LR.


**Concern #3**:

Does the improvement in performance justify the additional complexity of NQS++ (over Chinchilla)? In general, the reviewers noted that the complexity helped NQS perform better (Table 2) and made it possible to describe new dimensions like batch size, so we view this as a concern of potentially overfitting a complex model.

**Our Response**:

Our experiment design is aware of this concern: we use a training/validation/holdout split, the standard statistical approach to compare models with different levels of complexity. In our experiments, we found no evidence of NQS++ overfitting.

Because NQS++ is highly structured, it is not a particularly large model class; and as a single model, NQS is not more complex than Chinchilla augmented with additional methods to handle the batch size and time dimensions.

We acknowledge the limitation of using only one dataset (OWT2) and one model family (Pythia-like), which we hope to address in future works.

**Concern #4**:

The two modifications (EMS & LRA) we made over the vanilla quadratic model are empirically-motivated and do not seem well-justified from a mechanistic view point.

**Our Response**:

EMS is indeed heuristic-based; LRA, however, is highly mechanistic and corroborated by theory.

Although the implementation of LRA is new, we are not the first to propose an adaptive algorithm as a surrogate model for LLM training. McCandlish et al. (2018) suggests that a quadratic optimization model with line search can model LLM loss profiles. LRA can be viewed as a crude approximation of this line search. LRA modifies the “effective optimizer” of LLM training: such level of intervention with the training process is not possible with heuristic-based methods.

We added App. E.6 to show how LRA modified the training trajectory of NQS so that it closely matched that of LLMs.

### **List of New Figures and Tables**

- App. E.4: NQS++ performed well on a cosine LR schedule.
- App. E.5: Chinchilla failed to extrapolate over compute scales on its original Hoffman et al. dataset, consistent with our smaller experiments.
- App. E.6: LRA can be viewed as mechanistic, modifying the training trajectory of NQS to match that of LLMs.
- In progress: a sensitivity analysis (to the validation set, and EMS/LRA parameters).

### **Other Edits**

- Added theoretical motivation to LRA in Section 4.
- Added citation to Marek et al. 2025 in Appendix A.
- We aim to improve readability and flow in the next iteration of the draft.

---

> ### Author Response · Authors · 2025-11-27
> **New Experiment and Revised Draft**
>
> In addition to the new experiments in E.4, E.5, and E.6 (described above), we have now added:
>
> **New Experiment**
> - App. E.7: Sensitivity of the NQS++ fit to the EMS/LRA hyper-parameters, on the training, validation and test set.
>
> **Revision of the Draft**
> - Re-organising the introduction and preliminary sections to improve flow, including:
>     - a new background section (Sec. 2.2) that briefly reviews the theoretical scaling models, to prepare the readers for the definition of the NQS model class.

---

### Meta-Review · Area_Chair_2zTG · 2026-01-07

**Summary:**

Reviewers agree that they appreciate the combination of mechanical model and empirical method, and the introduction of batch size as an important factor for LLM training performance. Major concerns include

1. Limited model scale of the experiments (Reviewers 8D3C, GytH, maGC)
2. Unclear generalizability to different learning rates and hyperparameters (Reviewers 8D3C, GEC9)
3. Increased complexity of the model, which may defeat the purpose of scaling law (Reviewers 8D3C, GytH)
4. The empirical patches of NGS++ undermine the motivation of the theory-driven NQS model (Reviewers GytH, GEC9)
5. Unclear writing (Reviewers GytH, GEC9)
6. Lack of comparison with batch-size-aware scaling law baselines (Reviewer maGC)

**Reviewer Concerns:**

The authors are upfront about the limitations of the work, and acknowledged that some of them cannot be addressed with the current resources and timeframe, while addressing others. In particular -

1. The authors mentioned that they are seeking compute resources to conduct larger-scale experiments.
2. The authors conducted addtional experiments with cosine learning rate
3. The authors claimed that the added complexity of the proposed scaling law does not impact generalizability (within the limited scope); and released an easy-to-use package to remedy the increased complexity.
4. The authors acknowledged that the empirical patches is the bottleneck of the approach, and seek solutions as a future direction.
5. The authors updated the manuscript to improve the writing.
6. The authors pointed out that the batch-size-aware baselines do not share significant similarities with the proposed method so that are not directly comparable.

**Reviewer Scores:**

All the reviewers are likely to maintain their scores, because the limited model sizes and model families, as the most prominent weakness of the paper, cannot be resolved with the limited resources.

---

### Decision · Program_Chairs · 2026-01-26

Reject